# Outlier Weighed Layerwise Sparsity (OWL 👓): A Missing Secret Sauce for Pruning LLMs to High Sparsity

## Abstract

Large Language Models (LLMs), renowned for their remarkable performance across diverse domains, present a challenge due to their colossal model size when it comes to practical deployment. In response to this challenge, efforts have been directed toward the application of traditional network pruning techniques to LLMs, uncovering a massive number of parameters can be pruned in one-shot without hurting performance. Building upon insights gained from pre-LLM models, particularly BERT-level language models, prevailing LLM pruning strategies have consistently adhered to the practice of uniformly pruning all layers at equivalent sparsity levels, resulting in robust performance. However, this observation stands in contrast to the prevailing trends observed in the field of vision models, where non-uniform layerwise sparsity typically yields substantially improved results. To elucidate the underlying reasons for this disparity, we conduct a comprehensive analysis of the distribution of token features within LLMs. In doing so, we discover a strong correlation with the emergence of outliers, defined as features exhibiting significantly greater magnitudes compared to their counterparts in feature dimensions. Inspired by this finding, we introduce a novel LLM pruning methodology that incorporates a tailored set of **non-uniform layerwise sparsity ratios** specifically designed for LLM pruning, termed as **O**utlier **W**eighed **L**ayerwise sparsity (**OWL**). The sparsity ratio of OWL is directly proportional to the outlier ratio observed within each layer, facilitating a more effective alignment between layerwise weight sparsity and outlier ratios. Our empirical evaluation, conducted across the LLaMA-V1 family and OPT, spanning various benchmarks, demonstrates the distinct advantages offered by OWL over previous methods. For instance, our approach exhibits a remarkable performance gain, surpassing the state-of-the-art Wanda and SparseGPT by **61.22** and **6.80** perplexity at a high sparsity level of 70%, respectively. Code is submitted.

## 1 Introduction

The remarkable performance exhibited by Large Language Models (LLMs) across a diverse spectrum of applications has ignited an unparalleled race among tech giants and academic institutions to build LLMs at the billion-parameter scale (Brown et al., 2020; Touvron et al., 2023a;b; Brown et al., 2020). The compelling performance of LLMs demonstrated in various applications triggers an unprecedented competition of building billion-level LLMs among tech giants and academic institutions (Brown et al., 2020; Touvron et al., 2023a;b; Brown et al., 2020). While their exceptional capabilities are undeniable, the colossal size and computational demands of these models have also raised substantial concerns, particularly in terms of financial expenditure and environment (Luccioni et al., 2022; Patterson et al., 2021).

Network pruning (Mozer & Smolensky, 1989; Janowsky, 1989; LeCun et al., 1989; Han et al., 2015), as a long-established model compression method, is expected to serve as an effective solution for reducing the size of LLMs. However, network pruning usually favors a certain time of fine-tuning or re-training to reacquire the original optimal performance. Given the extensive text corpus and model size associated with LLMs, conventional fine-tuning becomes exceedingly challenging and less desirable. Fortunately, recent endeavors have explored the possibility of LLM pruning without the need for fine-tuning, showcasing that LLMs contain a substantial number of parameters that can

be removed in a single step with minimal performance degradation (Jaiswal et al., 2023; Frantar & Alistarh, 2023; Sun et al., 2023). SparseGPT (Frantar & Alistarh, 2023) addresses the challenge of LLM pruning from the perspective of layerwise reconstruction problem. In this context, the primary goal is to minimize the output discrepancy in terms of the reconstruction error between dense and sparse LLMs. It adopts an iterative strategy to handle the computational hurdle posed by the row-Hessian problem. Specifically, it employs the Optimal Brain Surgeon (OBS) algorithm (Hassibi et al., 1993) to selectively prune and update weights in a column-wise manner. Wanda (Sun et al., 2023), on the other hand, introduces a novel pruning metric that takes into account both the weight magnitudes and their corresponding input activations. Remarkably, it achieves performance on par with SparseGPT without relying on computationally expensive second-order information. The effectiveness of Wanda stems from the emergence of the outlier features residing within large-scale LLMs. These outliers, which tend to be significantly larger than typical features, are nonetheless crucial for optimizing LLM performance (Dettmers et al., 2022). In general, both SparseGPT and Wanda exhibit competitive performance, showcasing their ability to reduce model parameters by up to 50% while incurring only a modest increase of approximately 1 in perplexity (Sun et al., 2023).

It is worth noting that SparseGPT and Wanda unanimously follow previous work on BERT pruning (Sanh et al., 2020; Kurtic et al., 2022) and choose to prune LLMs with a uniform sparsity ratio per layer, *i.e.,* each layer will be pruned at the same sparsity. Such choice is reasonable for LLMs, as the pruning process typically involves sorting the importance scores of weights. Conducting such sorting globally across layers could become a computational bottleneck, especially for models at the billion-parameter scale. Nevertheless, before it has been taken root that uniform layerwise sparsity is the default choice for LLMs, we raise a timely inquiry: *are there any pivotal aspects that have been inadvertently omitted in the context of favorable layerwise sparsity ratios for LLM pruning?*

Three reasons behoove us to pose the above research question: *First*, it is widely acknowledged that within Transformer architectures, certain components hold greater significance than others, and thus, they merit distinct treatment during the pruning process (Wang & Tu, 2020; Bhojanapalli et al., 2021); *Second*, a consensus view has been reached in computer vision that non-uniform layerwise sparsity typically achieves stronger results than uniform sparsity (Liu et al., 2022; Lee et al., 2020); *More importantly*, LLMs demonstrate astonishingly emergent behaviors (Dettmers et al., 2022; Wei et al., 2022; Schaeffer et al., 2023) as model size continuously scales up, a phenomenon distinct from smaller-scale language models such as BERT (Devlin et al., 2018). These emergent behaviors offer fresh insights into the domain of LLM pruning. For instance, Dettmers et al. (2022) revealed the existence of outlier features within LLMs, with magnitudes up to 20 times larger than others, exerting a profound influence across all Transformer layers.

**Contributions.** Given the pivotal role that outliers play in the performance of LLMs, coupled with the demonstrated effectiveness of Wanda (Sun et al., 2023), our initial investigation centers on a systematic examination of the impact of existing LLM pruning methodologies on outliers. To our astonishment, we uncover a **compelling correlation** between pruning efficacy and the retention ratio of outliers: contemporary state-of-the-art LLM pruning approaches, such as SparseGPT and Wanda, exhibit remarkable preservation of outliers, even though the former was not originally designed with this intent. Moreover, we conduct an in-depth analysis of the distribution of outliers across different layers and observe **a notably non-uniform pattern**. This non-uniform distribution emerges as a valuable indicator for the formulation of layerwise sparsity strategies tailored specifically for LLMs. Building upon this newfound insight, we introduce an LLM pruning paradigm characterized by a novel layerwise sparsity ratio, denoted as **O**utlier **W**eighed **L**ayerwise sparsity (**OWL**). OWL inherently assigns greater emphasis to layers housing a higher prevalence of outliers, thereby facilitating more nuanced coordination between sparsity in weight matrices and the presence of outliers within the layer.

We conduct extensive experiments to evaluate the performance OWL across a spectrum of large language models, including LLaMA-V1 family (Touvron et al., 2023a), and OPT (Zhang et al., 2022), from 7B to 65B. Our empirical results show that OWL consistently outperforms existing top-performing LLM pruning methods, particularly at high sparsity levels. For instance, we observe significant improvements achieved by OWL over Wanda with LLaMa-7B on WikiText (Merity et al., 2016a), with perplexity reductions of more than 60 and 3300 perplexity at sparsity levels of 70% and 80%, respectively. Our research presents a compelling counter-argument to previous study by shedding light on the previously overlooked yet crucial role of layerwise sparsity ratios in the context of LLM pruning. This shift in perspective has allowed us to push the boundaries of achievable LLM pruning ratios to reach 70% without the need of any weight updates or second-order Hessian.

## 2 RELATED WORK

**Pruning and LLM Pruning.** Since the 1980s, network pruning has been a well-established technique for simplifying neural networks in various applications while maintaining accuracy (Mozer & Smolensky, 1989; Han et al., 2015; Mocanu et al., 2018; Wen et al., 2017; Lin et al., 2019). However, when it comes to pruning Large Language Models (LLMs), progress has been limited. Traditional pruning typically requires a round of re-training to restore performance, which can be challenging for LLMs. To address this challenge, researchers have developed pruning algorithms specifically tailored for LLM compression. For example, Ma et al. (2023) explored structured sparse LLMs using Taylor pruning to remove entire weight rows, followed by LoRA fine-tuning (Ma et al., 2023). Recent research has shifted toward unstructured pruning without the need for fine-tuning, showing substantial advancements. SparseGPT (Frantar & Alistarh, 2023) utilizes the Hessian inverse for pruning and with subsequent weight updates to reduce reconstruction error of dense and sparse weights, while Wanda (Sun et al., 2023) produces a criterion incorporating weight magnitude with their input activations, aiming to preserve outlier features (Dettmers et al., 2022). Our work for the first time probe and highlight the crucial role of non-uniform layerwise sparsity for LLM pruning, making a notable progress in this field.

**Layerwise Sparsity for Pruning.** While it is common to use uniform layerwise sparsity (Zhu & Gupta, 2017; Gale et al., 2019) to prune language models (Sanh et al., 2020; Kurtic et al., 2022), there is a well-established line of work that explore non-uniform layerwise sparsity in terms of pruning vision models. Mocanu et al. (2016) propose a non-uniform and scale-free topology inspired from graph theory, showing better performance than the dense counterpart when applied to restricted Boltzmann machines. Follow-up works significantly improve its scalability based on *Erdős-Rényi* graph (Erdős & Rényi, 1959), extending to fully-connected layers (Mocanu et al., 2018) and convolutional layers (Evci et al., 2020; Liu et al., 2022) as data-free and feedforward-free layerwise sparsity. Another group of work produces non-uniform sparsity by applying a global threshold on every layer (Frankle & Carbin, 2019; Lee et al., 2019; Wang et al., 2020; Lee et al., 2020; Liu et al., 2021). However, global pruning becomes extremely expensive and inefficacious in the context of LLM pruning as shown in Table 2. We also provide a comparison among most common layerwise sparsity for LLMs in Section 5, and all of them fail to perform on LLMs.

**Outliers in LLMs.** Unlike traditional vision or smaller-scale transformer models, recent studies have revealed certain emergent characteristics unique to language models at scale. Specifically, one intriguing trait of LLMs is the exhibition of *outlier features*, which are the features with significantly larger magnitudes than others (Dettmers et al., 2022). While constituting only a very small portion of the entire feature dimensions, these outliers play an imperative role in models' predictive performance. Building upon this observation, several recent works have developed techniques to effectively quantize LLMs with minimal performance drop (Dettmers et al., 2022; Xiao et al., 2023; Lin et al., 2023). On the other hand, in the context of LLM pruning, this unique characteristic has scarcely been taken into account to the best of our knowledge (Sun et al., 2023). Our work draws on the importance of the emergent outliers in LLMs, and provides a systematic study on its correlation to the effectiveness of model pruning, leading to a novel technique that leverages the distribution of outliers to guide layerwise LLM pruning.

## 3 OUTLIER WEIGHED LAYERWISE SPARSITY – OWL

In this section, we will introduce **O**utlier **W**eighed **L**ayer-wise sparsity (**OWL**) step by step, from rationales, to empirical studies, and eventually to the algorithm.

### 3.1 RATIONALE

The primary of goal of network pruning is to discover the least important components, such as individual weights in the case of unstructured pruning, which have minimal impact on the model's output. In the context of pre-LLMs with smaller scales, magnitude pruning has traditionally serves as the most basic yet effective technique, consistently delivering robust results across various scenarios (Han et al., 2015; Mocanu et al., 2018; Frankle & Carbin, 2019; Jaiswal et al., 2023). The effectiveness of magnitude pruning in compressing pre-LLM models is closely intertwined with the feasibility of fine-tuning. It has been observed that even the random removal of components can ultimately restore the original performance through adequate fine-tuning (Liu et al., 2022; Mittal et al., 2019). However, fine-tuning encounters significant challenges when applied to LLMs, rendering magnitude pruning less effective compared to more precise pruning metrics, such as second-order

Hessian (Frantar & Alistarh, 2023) and input activation (Sun et al., 2023). Notably, Wanda (Sun et al., 2023) achieves remarkable performance by augmenting input activation with weight magnitude, underscoring the critical importance of preserving outlier features in LLM pruning. Considering the vital role that outliers play in the context of LLMs (Dettmers et al., 2022) and the success of Wanda, we conjecture that the performance of different pruning methods has a strong correlation with their ability to preserve outlier features. To assess our conjecture, we undertake preliminary investigations outlined below based on Layerwise Outlier Distribution.

## 3.2 EMPIRICAL STUDY

**Layerwise Outlier Distribution (`LOD`).** Our preliminary studies are based on Layerwise Outlier Distribution (`LOD`), a concept used to measure how outlier features distribute and effect weights across layers. Since we focus on weight pruning in this paper, instead of measuring the outlier distribution of input features, We opt to prioritize the impact of outlier features on weights, which is quantified as the accumulation of all input features connected to the target weight, multiplied by the weight magnitude (Sun et al., 2023). Our intuition here is that weights that are most affected by outliers also play a pivotal role in propagating and preserving these outlier features.

To formalize our approach, we consider the input of a layer as $\mathbf{X}$ with dimensions $(N \times L, C_{\texttt{in}})$, where $N$ and $L$ represent the batch and sequence dimensions, respectively. The weight matrix $\mathbf{W}$ has dimensions $(C_{\texttt{out}}, C_{\texttt{in}})$. The impact of input features $\mathbf{X}$ on weight $\mathbf{W}_{\texttt{ij}}$ is computed as $\mathbf{A}_{\texttt{ij}} = \|\mathbf{X}_{\texttt{j}}\|_2 \cdot |\mathbf{W}_{\texttt{ij}}|$, which is the aggregation of all input features connected to weight $\mathbf{W}_{\texttt{ij}}$, multiplied by its magnitude $|\mathbf{W}_{\texttt{ij}}|$. Here, $\|\mathbf{X}_{\texttt{j}}\|_2$ is the $\ell_2$ norm of the $j^{th}$ feature of input $\mathbf{X}$. This computation is performed across all $N \times L$ tokens, resulting in a scalar value denoted as $\|\mathbf{X}_{\texttt{j}}\|_2$. It is worth noting that $\mathbf{A}_{\texttt{ij}}$ also serves as the pruning metric used by Wanda (Sun et al., 2023) to assess the importance of weight $\mathbf{W}_{\texttt{ij}}$. Subsequently, after obtaining the impact of features for all weights $\mathbf{A}$, we proceed to calculate the "outlier ratio" of $\mathbf{A}$ by identifying elements whose magnitude is $\mathbf{M}$ times greater than the averaged value in each layer. We empirically find that both $\mathbf{M} = 5$ or $\mathbf{M} = 7$ effectively sketch the distribution of the impact of outliers features on weights. This process enables us to derive a vector, denoted as `LOD` $= [D_1, D_2, ..., D_l]$, which characterizes the layerwise outlier distribution *w.r.t.,* the impact of features on weights within a $l$-layer LLMs. Formally, the definition of `LOD` is given as:

$$\texttt{LOD} = \frac{\sum_{i=1}^{C_{\texttt{out}}} \sum_{j=1}^{C_{\texttt{in}}} \mathbb{I}(\mathbf{A}_{\texttt{ij}} > \text{mean}(\mathbf{A}) \times \mathbf{M})}{C_{\texttt{in}} C_{\texttt{out}}}$$

where $\mathbb{I}(\cdot)$ is the indicator function, which returns 1 when the condition is satisfied. Based on `LOD`, we conduct three empirical studies outlined below to better understand LLM pruning.

**Empirical Study I: Dense LLMs vs. `LOD`.** To investigate whether sparsifying LLMs necessitates differential treatment of individual layers, we employ `LOD` to gauge the layerwise distribution of outliers within dense LLMs. If `LOD` in dense LLMs exhibits a relatively uniform pattern, it suggests that a non-uniform layerwise distribution may not be imperative, at least in terms of outlier features, and vice versa. We assess the `LOD` across various dense LLMs, including LLaMA-7B, 13B, and 30B.

**Empirical Study II: Pruning Metric vs. `LOD`.** We further delve into the impact of different pruning metrics on `LOD`. The primary objective of this study is to explore whether there exists a robust correlation between the performance of various pruning methods and their ability to preserve outliers. To achieve this, we *aggregate* the `LOD` values across layers for various LLM pruning methods, including magnitude, Wanda, and SparseGPT, and compare them with their dense counterparts. In order to mitigate the influence of pruning on the average value of $\mathbf{A}$, we maintain consistency by utilizing the pre-pruning average value to measure the outlier ratio after pruning. Subsequently, the number of outliers after pruning is then divided by the total number of weights in the layer (including both zero and non-zero weights) to obtain the updated outlier ratio after pruning. Doing so helps avoid the impact of pruning on average values, ensuring a precise evaluation of alterations in the outlier ratio. All sparse models are pruned with uniform layerwise sparsity. These experiments are conducted using LLaMA-13B at sparsity level of 60% and 70% with $\mathbf{M} = 7$.

**Empirical Study III: Pruning Granularity.** It is well-established that non-uniform or global layerwise sparsity often leads to more accurate sparser networks at high sparsity than the uniform layerwise sparsity for pre-LLM pruning. However, endeavors unanimously point out that uniform sparsity is more favorable when pruning LLMs (Frantar & Alistarh, 2023; Sun et al., 2023). To gain

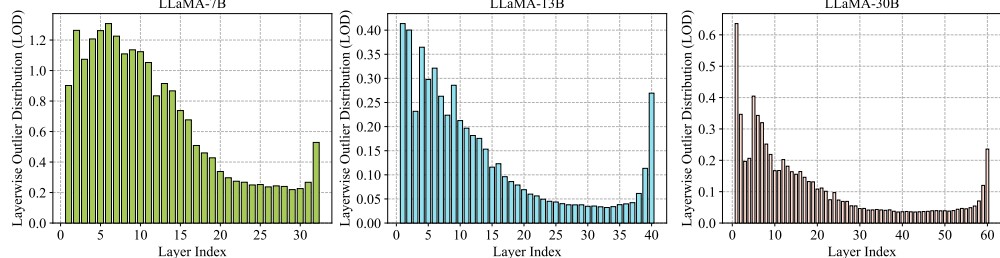

Figure 1: Layerwise Outlier Distribution (LOD) (%) of dense LLaMA-7B, 13B, and 30B.

deeper insights into these seemingly contradictory arguments, we conducted a study to systematically investigate the impact of different pruning granularities on LLM pruning. Specifically, we study two sets of pruning granularities: (1) **Across different layers,** we compare the performance of uniform sparsity and global sparsity; (2) **Within the same layer,** we study the output-imbalanced sparsity used by SparseGPT against the output-balanced sparsity adopted by Wanda. Output-balanced sparsity eliminates the same amount of weights for all outputs. We conduct experiments with magnitude pruning and Wanda using LLaMA-7B at various sparsity.

**Results:** We present our findings from Study 1-3, in Figure 1, Table 1, and Table 2, respectively. These results provide positive support for our conjecture, and we summarize the key observations below:

① **LOD of dense LLMs exhibits a highly non-uniform distribution across layers.** In essence, the distribution of dense LLMs shown in Figure 1 loosely follows a "U" shape, with notable proportions at both ends, while the central region displays a monotonic descending trend. This finding validates our conjecture that individual layers need unique consideration during the pruning procedure. Employing uniform pruning across all layers would inevitably disrupt the outlier structure in layers characterized by a large outlier ratio, such as those layers at the beginning or end of models.

Table 1: Effects of various pruning methods on Layerwise Outlier Distribution (LOD) and Perplexity with LLaMA-13B on WikiText. LOD is calculated as the *summation* across all layers with $\mathbf{M} = 7$.

| Sparsity | Method | LOD (%) ↑ | ΔLOD (%) ↑ | Perplexity ↓ |
|---|---|---|---|---|
| | Dense | 5.432 | - | 5.090 |
| 70% | Wanda | 5.716 | 0.284 | 55.900 |
| | SparseGPT | **6.645** | **1.213** | **19.235** |
| | Magnitude | 5.322 | -0.110 | 84539.445 |
| 60% | Wanda | 5.433 | 0.001 | 8.761 |
| | SparseGPT | **6.044** | **0.612** | **8.458** |
| | Magnitude | 5.322 | -0.110 | 229.451 |

② **The performance of sparse pruning methods on LLMs is closely correlated with their ability to retain outlier features.** Leading pruning techniques like Wanda and SparseGPT all excel in outlier, resulting in an overall increase in LOD. In contrast, the naive baseline of magnitude pruning performs no better than random selection at 70% sparsity, as evidenced by a negative change of -0.110 in LOD, indicating the removal of important outliers. It is interesting to see that despite SparseGPT not being explicitly designed for outlier preservation, it achieves the highest LOD as well as performance, providing further insight into the underlying reason for its success. A plausible reason is that the weight update involved within SparseGPT helps increase LOD.

Table 2: WikiText perplexity with LLaMA-7B of various pruning granularity.

| Method | Layerwise Uniform | Output Balanced | 10% | 20% | 30% | 40% | 50% | 60% | 70% |
|---|---|---|---|---|---|---|---|---|---|
| Wanda | ✓ | ✓ | 5.697 | 5.817 | 5.999 | 6.388 | 7.260 | 10 | 86 |
| Wanda | ✓ | ✗ | 5.695 | 5.819 | 6.029 | 6.572 | 7.942 | 20 | 238 |
| Wanda | ✗ | ✗ | 14.117 | 3134 | 10293 | 10762 | 14848 | 17765 | 5147 |
| Magnitude | ✓ | ✓ | 5.803 | 6.018 | 6.622 | 8.041 | 13.349 | 152 | 25304 |
| Magnitude | ✓ | ✗ | 5.806 | 6.020 | 6.669 | 8.601 | 17.287 | 559 | 48419 |
| Magnitude | ✗ | ✗ | 5.821 | 6.111 | 7.012 | 9.825 | 48.627 | 38335 | 29283 |

③ **Pruning with coarser granularity results in diminished performance.** In general, we observe a consistent trend of improved perplexity as the pruning granularity becomes finer, transitioning from global layerwise sparsity to uniform layerwise sparsity at the macro level, and from output-imbalanced sparsity to output-balanced sparsity at the micro level. These findings align with the conclusions presented by Sun et al. (2023). One plausible explanation for this trend is that coarser-grained pruning tends to eliminate more outlier features, particularly in certain layers or outputs.

### 3.3 OUTLIER WEIGHED LAYERWISE SPARSITY (OWL)

The above empirical studies underscore the critical significance of preserving outliers in the context of LLM pruning. Consequently, it becomes imperative to implement layerwise pruning strategies that take into account the non-uniform distribution of outliers across different layers. However, global pruning can be costly and lead to collapse of outliers, resulting in significant performance degradation. On the other hand, uniform pruning does not adequately consider the highly non-uniform distribution of outlier features across various layers. This negligence inevitably disrupts the structure of outliers in layers characterized by a substantial outlier ratio, particularly at high sparsity levels. Therefore, there is a need of an ideal layerwise sparsity that aligns effectively with the layerwise outlier distribution while maintaining computational and memory efficiency.

To address this issue, we propose a novel layerwise sparsity ratio strategy, referred to as **O**utlier **W**eighed **L**ayer-wise sparsity (**OWL**) explicitly tailored for Large Language Models, which can better coordinate with the outlier distribution by taking the layerwise outlier ratio into consideration. Given a $l$-layer large language model with a target model sparsity $S$, we aim to calculate the target layerwise sparsity $[S_1, S_2, ..., S_n]$. We first calculate LOD of feature effects on weights, $\mathbf{D} = [D_1, D_2, ..., D_n]$, based on the approach proposed in Section 3.2. Guided by the principle that layers with a higher proportion of outliers should have a lower sparsity, we set $S_i \propto 1 - D_i$. Additionally, we introduce a hyperparameter $\lambda$ which constrains the layerwise sparsity to fall within a specified range, specifically, $S_i \in [S - \lambda, S + \lambda]$, while maintaining an average sparsity of $S$ across all layers. This helps prevent excessive difference in sparsity between layers, ensuring a robust performance. This constraint is inspired by the insights gained from "Empirical Study III" which highlight the detrimental impact of overly aggressive layerwise sparsity, akin to global pruning, on sparse LLMs. To obtain a favorable number for $\lambda$ and $M$, we conduct a small hyperparameter sweep within the range of $\lambda \in [0.02, 0.05, 0.08, 0.1, 0.2]$ and for $M \in [3, 5, 7, 10]$. The visualization of our layerwise sparsity ratio is demonstrated in Figure 2, where we can clearly see that the layerwise sparsity level of OWL nuancedly aligns with model's LOD.

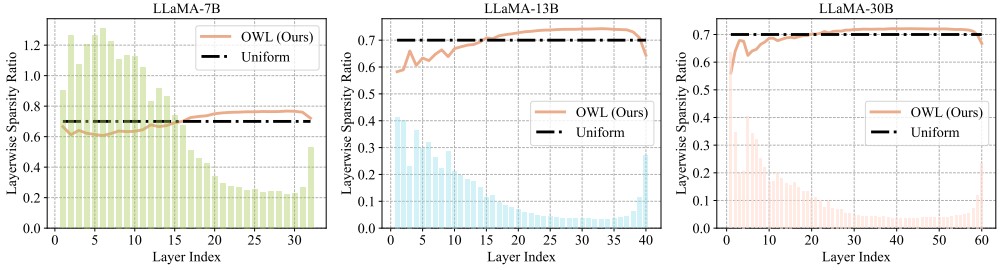

Figure 2: The demonstration of the OWL layerwise sparsity and Uniform layerwise sparsity at 70% sparsity. The bar chart in background corresponds to the Layerwise Outlier Distribution (LOD).

## 4 EXPERIMENTS

**Models and Dataset.** We assess OWL's performance across a range of LLMs, encompassing the LLaMA-V1 model family (Touvron et al., 2023b) with parameter counts ranging from 7 billion to 65 billion, as well as OPT-6.7B (Zhang et al., 2022). Our evaluation protocol aligns with established LLM pruning methodologies (Frantar & Alistarh, 2023; Sun et al., 2023), encompassing assessments of language modeling proficiency and zero-shot capabilities of sparse LLMs. Specifically, we measure the Perplexity metric on the WikiText (Merity et al., 2016b) validation dataset for language modeling performance, and employ the Accuracy metric for zero-shot evaluations on seven common sense benchmarks, including BoolQ (Clark et al., 2019), RTE (Wang et al., 2018), HellaSwag (Zellers

et al., 2019), WinoGrande (Sakaguchi et al., 2019), ARC Easy and Challenge (Clark et al., 2018), and OpenbookQA (Mihaylov et al., 2018).

**Baselines.** We choose the three current LLM-pruning baselines, including magnitude (Jaiswal et al., 2023), SparseGPT (Frantar & Alistarh, 2023), Wanda (Sun et al., 2023). Magnitude pruning serves as a naive baseline for LLMs, with an expected sharp decline in performance at modest sparsity levels, typically ranging from 10% to 30%. SparseGPT and Wanda, on the other hand, are established baselines known for their ability to maintain reasonable performance even at relatively high sparsity levels, typically around 50% to 60%. Notably, in contrast to our approach, all baseline methods employ with uniform layerwise sparsity. We primarily focus on high sparsity levels, not falling below 50%, as regions with low sparsity pose challenges for existing sparse GPU kernels to outperform their dense counterparts (Gale et al., 2020). To ensure equitable comparisons, we have employed the identical set of calibration data as utilized by SparseGPT and Wanda for model pruning, *i.e.,* comprising 128 sequences with 2048 tokens for each, randomly sampled from the first shard of the C4 (Raffel et al., 2020) dataset. We incorporate OWL directly into Wanda and SparseGPT, resulting in two variants: "OWL *w.* Wanda" and "OWL *w.* SparseGPT". The only distinction between these variants lies in their layerwise sparsity ratios, with OWL providing a more tailored layerwise sparsity in this regard. Hyperparameters are shared in Table 4-Right.

Table 3: WikiText validation perplexity of pruning methods for LLaMA-V1 family and OPT-6.7B at 70% sparsity. The best performance method is indicated in **bold**, and the gain in perplexity achieved by OWL is highlighted in blue.

| Method | Layerwise Sparsity | Weight Update | LLaMA-V1 | | | | OPT 6.7B |
| --- | --- | --- | --- | --- | --- | --- | --- |
| | | | 7B | 13B | 30B | 65B | 6.7B |
| Dense | - | - | 5.68 | 5.09 | 4.10 | 4.77 | 10.13 |
| Magnitude | Uniform | ✗ | 48419.12 | 84539.45 | 977.73 | 46.89 | 290985.03 |
| Wanda | Uniform | ✗ | 85.77 | 55.90 | 17.37 | 15.23 | 162.92 |
| OWL *w.* Wanda | Non-Uni | ✗ | 24.55 (-61.22) | 17.17 (-38.73) | 10.75 (-6.62) | 8.61 (-6.62) | 40.22 (-120.70) |
| SparseGPT | Uniform | ✓ | 26.30 | 19.24 | 12.56 | 10.45 | 20.29 |
| OWL *w.* SparseGPT | Non-Uni | ✓ | 19.49 (-6.81) | 14.55 (-4.69) | 10.28 (-2.28) | 8.28 (-0.64) | 22.48 (2.19) |

## 4.1 EXPERIMENTAL RESULTS

**Language Modelling.** We first report the performance of various LLM pruning methods on language modelling with WikiText. The results is presented in Table 3 and Figure 3. We summarize the key observation below:

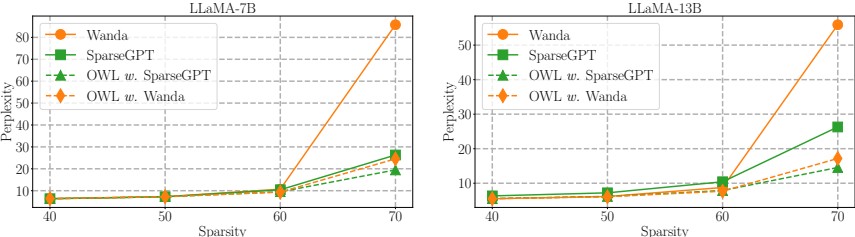

Figure 3: WikiText validation perplexity of OWL applied to SparseGPT and Wanda.

① **OWL demonstrates its versatility serving as a general layerwise sparsity method suitable for various scenarios.** As illustrated in Table 3, OWL exhibits effectiveness across different pruning methods (such as Wanda and SparseGPT), architectural variants (including LLaMA-V1 and OPT), and diverse model sizes (ranging from LLaMA-V1 with 7B, 13B, 30B, to 65B parameters), resulting in substantial reductions in perplexity scores. Notably, even when applied to SparseGPT, a strong pruning method incorporating second-order information, OWL still achieves significant perplexity reductions, exemplified by a reduction of 6.81 for LLaMA-7B.

② **The benefits of OWL increases as significantly model size decreases.** There is a clear trend that the performance gain of OWL monotonically increases as LLaMA-V1 scales down from 65B to 7B. While the performance improvement of OWL *.w* Wanda for LLaMA-65B is relatively small, at 6.62, it achieves a remarkable gain of 61.22 for LLaMA-7B, resulting in a reasonable 24.55 perplexity.

**Zero-Shot Tasks.** While perplexity is a widely used metric for language modeling, it primarily serves as a statistical measure of how confidently a language model predicts a text sample and does not necessarily align with the quality of the generated text. To draw more robust conclusions, we conducted experiments to evaluate the zero-shot ability of various sparse LLMs on diverse zero-shot downstream tasks with prompting. These experiments were performed using the LLaMA-V1 family at 70% sparsity, and the results are presented in Table 4. It's noteworthy that OWL consistently improves accuracy across nearly all settings, with very few exceptions on RTE data, which is . For example, OWL achieves an average perplexity gain of 4.72 and 2.19 over 7 tasks and 4 model sizes compared to Wanda and SparseGPT alone, respectively. This result highlights the promise of OWL is still hold for more challenging zero-shot downstream tasks.

Table 4: Accuracies (%) for 7 zero-shot tasks with 70% sparsity using LLaMA-V1 family.

| Params | Method | BoolQ | RTE | HellaSwag | WinoGrande | ARC-e | ARC-c | OBQA | Mean |
|---|---|---|---|---|---|---|---|---|---|
| | Dense | 75.14 | 66.43 | 74.80 | 70.01 | 67.67 | 41.38 | 41.40 | 62.40 |
| 7B | Magnitude | 38.29 | 52.71 | 24.68 | 51.46 | 26.98 | 22.35 | 25.80 | 34.61 |
| | Wanda | 55.11 | 57.40 | 31.83 | 51.38 | 34.22 | 19.80 | 26.00 | 39.39 |
| | OWL *w.* Wanda | **62.48** | **58.48** | **44.79** | **58.72** | **45.03** | **26.19** | **29.60** | **46.47** |
| | SparseGPT | 64.53 | **53.79** | 42.11 | 58.64 | 43.06 | 24.57 | 27.80 | 44.93 |
| | OWL *w.* SparseGPT | **67.13** | 53.43 | **48.56** | **62.03** | **45.41** | **27.65** | **32.00** | **48.03** |
| | Dense | 77.86 | 70.40 | 78.08 | 72.77 | 69.19 | 47.18 | 43.80 | 65.61 |
| 13B | Magnitude | 52.94 | 50.54 | 27.67 | 50.91 | 28.24 | 23.38 | 24.80 | 36.93 |
| | Wanda | 61.71 | **52.71** | 34.31 | 52.33 | 37.16 | 20.90 | 29.60 | 41.25 |
| | OWL *w.* Wanda | **62.69** | **52.71** | **51.03** | **63.14** | **49.54** | **28.67** | **34.40** | **48.88** |
| | SparseGPT | **66.94** | 52.71 | 47.91 | 62.90 | 45.03 | 27.99 | 35.20 | 48.38 |
| | OWL *w.* SparseGPT | 64.95 | **53.07** | **54.39** | **66.54** | **48.86** | **30.12** | **38.00** | **50.85** |
| | Dense | 82.69 | 66.79 | 81.19 | 75.85 | 73.48 | 50.77 | 44.60 | 67.91 |
| 30B | Magnitude | 39.14 | 46.21 | 24.31 | 52.33 | 24.66 | 22.87 | 29.00 | 34.07 |
| | Wanda | 66.12 | **57.76** | 58.84 | 67.32 | 59.26 | 33.11 | **40.20** | 54.66 |
| | OWL *w.* Wanda | **66.42** | 52.35 | **62.94** | **69.30** | **61.83** | **35.84** | 40.00 | **55.53** |
| | SparseGPT | 66.51 | **63.90** | 60.38 | 69.85 | 58.54 | 33.70 | 40.60 | 55.78 |
| | OWL *w.* SparseGPT | **67.58** | 58.48 | **64.88** | **70.72** | **60.82** | **35.07** | **42.20** | **57.11** |
| | Dense | 84.86 | 69.68 | 82.94 | 77.35 | 75.08 | 52.56 | 44.20 | 69.52 |
| 65B | Magnitude | 52.17 | 54.87 | 49.87 | 56.67 | 49.71 | 30.63 | 38.80 | 47.53 |
| | Wanda | 76.30 | 56.68 | 61.26 | 70.48 | 63.47 | 35.67 | 39.40 | 57.61 |
| | OWL *w.* Wanda | **80.12** | **58.84** | **66.16** | **73.56** | **65.45** | **39.93** | **42.20** | **60.89** |
| | SparseGPT | 80.64 | 59.57 | 66.42 | 72.61 | **60.52** | 38.57 | **40.80** | 59.88 |
| | OWL *w.* SparseGPT | **82.63** | **67.15** | **68.52** | **75.06** | 60.10 | **39.59** | 39.00 | **61.72** |

## 5 ANALYSIS

### 5.1 COMPARISONS AMONG VARIOUS LAYERWISE SPARSITY

We compare OWL layerwise sparsity with multiple commonly used layerwise sparsity, including:

- **Global** (Frankle & Carbin, 2019). A global threshold is uniformly applied to all layers to satisfy the overall sparsity requirement, and the specific layerwise sparsity is automatically adjusted based on this threshold.
- **Uniform** (Zhu & Gupta, 2017). Every layer is pruned with the same target sparsity.
- **Erdős-Rényi (ER)** (Mocanu et al., 2018). The sparsity of the convolutional layer is scaled proportional to $1 - \frac{n^{l-1}+n^l}{n^{l-1} \times n^l}$ where $n^l$ refers to the number of neurons/channels in layer $l$.
- **ER-Plus** (Liu et al., 2022). ER-Plus modifies ER by forcing the last layer as dense if it is not, while keeping the overall parameter count the same.
- **OWL-inverse**. OWL-inverse metric is the inverse variant of OWL, whose outlier ratio is $1 - \texttt{LOD}$.

For this study, we apply Wanda to the LLaMA-7B model. The results are presented in Table 5. It is noteworthy that all approaches, except for the Global method, perform satisfactorily when the sparsity level is at or below 40%. This observation suggests that the region of low sparsity does not

provide significant distinctions for performance comparison. However, as the sparsity level exceeds 50%, discrepancies between the various approaches become evident. Notably, the Uniform and OWL methods emerge as the top-performing approaches, with OWL consistently outperforming the former across all sparsity levels. On the other hand, the ER family of methods appears to be less suitable for LLM pruning. It's worth mentioning that the performance of OWL experiences a significant decline when we invert its outlier ratio, underscoring the effectiveness of LOD in identifying critical layers.

Table 5: WikiText validation perplexity of LLaMA-7B with various layerwise sparsity using Wanda.

| Sparsity/Perplexity | 10% | 20% | 30% | 40% | 50% | 60% | 70% | 80% |
|---|---|---|---|---|---|---|---|---|
| Global | 14.11 | 3134 | 10293 | 10762 | 14848 | 17765 | 5147 | 39918.56 |
| ER-Plus | 5.70 | 5.82 | 6.05 | 6.62 | 8.00 | 14.04 | 229.17 | 6013.91 |
| ER | 5.69 | 5.80 | 6.02 | 6.55 | 7.74 | 12.16 | 112.03 | 11151.18 |
| Uniform | 5.69 | 5.81 | 5.99 | 6.38 | 7.26 | 10.70 | 85.77 | 3499.88 |
| OWL-inverse | 5.72 | 5.83 | 6.04 | 6.51 | 8.03 | 26.05 | 822.23 | 9616.08 |
| OWL (ours) | 5.70 | 5.80 | 6.01 | 6.39 | 7.22 | 9.35 | 24.54 | 1002.87 |

## 5.2 PRUNING EFFICIENCY

| | LLaMA | | | |
|---|---|---|---|---|
| Method | 7B | 13B | 30B | 65B |
| SparseGPT | 208 | 341 | 731 | 1297 |
| OWL *w.* SparseGPT | 208 | 342 | 733 | 1301 |
| Wanda | 0.3 | 0.6 | 1.1 | 1.8 |
| OWL *w.* Wanda | 0.5 | 1.3 | 2.0 | 3.7 |

| Model | M | $\lambda$ |
|---|---|---|
| LLaMA-7B | 5 | 8% |
| LLaMA-13B | 7 | 8% |
| LLaMA-30B | 5 | 8% |
| LLaMA-65B | 5 | 20% |
| OPT-6.7B | 10 | 8% |

Figure 4: **Left:** Comparison of time overhead (in seconds), excluding the shared forward pass process. **Right:** Hyperparameters used to reproduce the results in this paper.

Since we utilize the pruning metric of Wanda to determine our layerwise sparsity, the theoretical computational complexity of OWL is comparable to that of Wanda, which is expected to be significantly lower than SparseGPT. To demonstrate this, we measure the total pruning time, excluding the forward pass process, following the methodology outlined by Sun et al. (2023). These results were obtained using NVIDIA A100 GPUs.

Our results in Table 4 indicate that OWL introduces nearly negligible overhead when compared to SparseGPT. Conversely, OWL *.w* Wanda doubles the pruning time in comparison to Wanda alone, yet it efficiently prunes a 65B LLaMA model within only 4 seconds. This additional time overhead primarily arises from the computation of $\|\mathbf{X}_j\|_2 \cdot |\mathbf{W}_{ij}|$ for the computation of Layerwise Outlier Distribution (LOD). However, as Wanda also employs this metric for pruning, we believe there is potential for solutions to mitigate this overhead. This aspect is left for future work and further optimization.

## 6 EXPLORING MORE PRACTICAL USAGE OF OWL

While unstructured sparsity receives limited support on GPUs, it's worth noting that OWL holds significant potential in hardware-friendly scenarios. We explore the benefits of OWL in three more practical regimes: N:M sparsity, structured pruning, and mixed-precision quantization in Appendix 7.

## 7 CONCLUSION

In this paper, we focus on a crucial aspect of LLM pruning that have been overlooked by previous works – layerwise sparsity ratios. Despite the prevailing practice of uniformly pruning all layers at equivalent sparsity levels, as observed in prominent LLM pruning papers, our investigation diverges from this trend by drawing inspiration from the emergence of outliers, characterized by features exhibiting significantly greater magnitudes compared to others. Leveraging this discovery, we introduced a novel layerwise sparsity ratio known as Outlier Weighed Layerwise sparsity (OWL). OWL employs tailored non-uniform layerwise sparsity ratios designed specifically for LLM pruning, aligning sparsity ratios with outlier ratios within each layer. Notably, our approach demonstrates substantial performance gains, surpassing the state-of-the-art Wanda and SparseGPT by 61.22 and 6.80 perplexity points, respectively, at a high sparsity level of 70%. Our findings offer fresh insights into the critical significance of layerwise sparsity in the context of LLM pruning. This work opens

up new avenues for the development of specialized sparse algorithms that can further optimize the deployment of LLMs in practical applications.

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

# A EXPLORING MORE PRACTICAL USAGE OF OWL

In order to examine if the promise of our non-uniform layerwise sparsity holds in hardware-friendly regimes. We explore OWL in three hardware-friendly regimes, including N:M sparsity, structured pruning, and mixed-precision quantization.

## A.1 N:M SPARSITY

Following DominoSearch (Sun et al., 2021), we opt for mixed N:8 sparsity configuration. Instead of employing a uniform N value across all layers, we allows for individual layers to possess distinct N values while maintaining the overall parameter count **same**. We adopt OWL to determine the optimal value of N for individual layers. The results are reported in Table 6. We can see OWL achieves consistent performance improvement over uniform N:M sparsity. Notably, in high sparsity scenarios like 3:8 and 2:8 sparsity, OWL demonstrates a significant improvement with $2\times$ and $8\times$ perplexity reductions over the uniform baseline, respectively.

Table 6: Perplexity of mixed N:M sparsity (N refers to non-zero weights) with LLaMA-7B on WikiText.

| Method | Laywewise Sparsity | Structure | Perplexity |
|--------|--------------------|-----------|------------|
| Wanda | Uniform | 4:8 | 8.57 |
| Wanda | OWL | Mixed 4:8 | **8.55** |
| Wanda | Uniform | 3:8 | 42.56 |
| Wanda | OWL | Mixed 3:8 | **21.49** |
| Wanda | Uniform | 2:8 | 2962.00 |
| Wanda | OWL | Mixed 2:8 | **331.37** |

## A.2 STRUCTURED PRUNING

Instead of pruning individual weights, structured pruning involves the selective removal of an entire group of weights, which are more amenable to hardware speedup, including weight blocks, neurons, filters/channels, and attention heads (Liu & Wang, 2023). We follow the recent methodology introduced in LLM Pruner (Ma et al., 2023), wherein entire neurons and attention heads are removed. This action facilitates direct acceleration of pruned LLMs on GPUs or TPUs. We replace the uniform layerwise sparsity used by LLM pruner to the non-uniform layerwise sparsity discovered by OWL. Table 7 again shows that OWL achieves preferable performance compared to the uniform layerwise sparsity in the context of structured pruning.

Table 7: Perplexity of Structure Pruning with LLaMA-7B on WikiText and PTB.

| Dataset | Pruning Method | Layerwise Sparsity | 20% | 40% | 60% | 80% |
|---------|----------------|--------------------|-----|-----|-----|-----|
| WikiText | LLM Pruner | Uniform | 19.09 | 30.39 | 90.02 | 1228.17 |
| | LLM Pruner | OWL | **18.57** | **28.65** | **76.99** | **321.64** |
| PTB | LLM Pruner | Uniform | 29.51 | 66.90 | 192.06 | 1691.87 |
| | LLM Pruner | OWL | **28.82** | **53.22** | **150.16** | **502.07** |

Table 8: Perplexity of mixed-precision quantization with LLaMA-7B on WikiText.

| Method | Precision | Perplexity |
|---|---|---|
| Same Bit-width | 2 Bit | 104151.84 |
| Same Bit-width | 3 Bit | 25.82 |
| Same Bit-width | 4 Bit | 6.29 |
| Select with random | Mixed 3/4 Bit | 12.04 |
| Select with $L_1$ norm | Mixed 3/4 Bit | 14.61 |
| Select with OWL | Mixed 3/4 Bit | **9.09** |
| Select with random | Mixed 2/3/4 Bit | 11455.54 |
| Select with $L_1$ norm | Mixed 2/3/4 Bit | 13959.422 |
| Select with OWL | Mixed 2/3/4 Bit | **190.28** |
| Select with random | Mixed 2/4 Bit | 14817.12 |
| Select with $L_1$ norm | Mixed 2/4 Bit | 33670.214 |
| Select with OWL | Mixed 2/4 Bit | **7505.60** |

## A.3 MIXED-PRECISION QUANTIZATION

Leveraging our non-uniform layerwise sparsity, we can also enhance mixed-precision quantization by assigning higher precision to layers exhibiting more outliers. Following the approach outlined in (Tang et al., 2022), we utilize OWL to assign different bit precision to different layers, thereby facilitating a mixed-precision quantization strategy. Our baseline here is selecting with random and $L_1$ norm of weights. We can clearly see that OWL also functions as a good indicator to select important layers for mixed-precision quantization, facilitating better quantization performance.

