# OpenReview forum: "Outlier Weighed Layerwise Sparsity (OWL): A Missing Secret Sauce for Pruning LLMs to High Sparsity"
_ICLR.cc/2024/Conference — Submitted to ICLR 2024_

### Official Review · Reviewer_njDu · 2023-10-30

**Soundness:** 4 excellent
**Presentation:** 4 excellent
**Contribution:** 3 good
**Rating:** 8
**Confidence:** 5

**Summary:**

This paper proposed a method for assigning non-uniform layer-wise sparsity for pruning large language models. Previous works of SparseGPT and Wanda all adopt a layer-wise uniform strategies. It is thus possible to compress LLMs more with a non-uniform layerwise sparsity level. The proposed method, formally called Outlier Weighted Layerwise (OWL), assigned sparsity level per layer proportional to the outlier ratio in each layer. Experiment results demonstrated the effectiveness of the proposed method.

**Strengths:**

1. Non-uniform sparsity is an important problem in network pruning. Existing works use the same sparsity level across all layers. Gaining insight into which layer should be pruned more would be valuable for the pruning community.
2. The analysis on the outlier distribution is thorough and provide a clear motivation as to the proposed method OWL.
3. The proposed method OWL is not computationally extensive and could be applied easily on top of existing LLM pruning methods.
4. The finding that the effectiveness of existing pruning methods is closely related to the ability to retain outlier features is interesting.
5. Empirical results are strong, which outperform both Wanda and SparseGPT by a large margin on high sparsity levels.

**Weaknesses:**

1. The analysis of this paper on non-uniform sparsity is mostly focused on the whole Transformer block. However, less insights are provided as to which type of layer should be pruned more.
2. Unstructured sparsity may be less practical as it needs special hardware for acceleration.

**Questions:**

1. It would be good to discuss how the approach could be applied to non-uniform sparsity per FC layer rather than the entire Transformer block. In this way, we could understand better which type of layers in Transformer should be pruned more.
2. Could the definition of LOD in section 3.2. be defined more precisely? This seems to be an important analysis target in section 3. Currently it is described by text which is somewhat hard to understand.
3. While it is not the focus of this work, I am interesting as to how would OWL perform on other networks, e.g. ConvNet or Vision Transformer. Adding such analysis would be beneficial as to understand the effect of outliers on pruning LLMs.
4. While structured sparsity enforces a fixed sparsity ratio 50%, i think OWL could be potentially applied for this type of sparsity. Instead of a dynamic sparsity ratio, we could choose to either prune a FC layer with 2:4/4:8 sparsity or skip this FC layer entirely. I think such analysis would make this work more comprehensive.

---

> ### Author Response · Authors · 2023-11-20
> **Response to Reviewer  njDu (1/3)**
>
> We sincerely appreciate your positive score. We are glad that you found our paper valuable for the pruning community, our analysis on outlier distribution is thorough and provides a clear motivation to OWL, and empirical results are strong. We would like to address your concerns below.
>
> **Q1:** Less insights are provided as to which type of layer should be pruned more.
>
>
> - We thank you for raising this great question. As you suggested, we explored OWL in a per-layer basis. The comparison of per-layer and per-Transformer block is reported in the following tables. We report the sparsity ratio of 7 FC layers including q\_proj, k\_proj, v\_proj, o\_proj, gate\_proj, down\_proj, and up\_proj of layer 1, 2, 15, 30, and 31.
>
>
>     *Table: OWL per-layer. 70% Sparsity, 86.285 Perplexity*
>
>   | Layer | q\_proj | k\_proj | v\_proj | o\_proj | gate\_proj | down\_proj | up\_proj |
>   |-------|---------|---------|---------|---------|------------|------------|----------|
>   | 1     | 0.639   | 0.638   | **0.691** | 0.598   | **0.710**   | 0.696      | **0.713** |
>   | 2     | 0.680   | 0.677   | **0.707** | 0.679   | **0.711**   | 0.703      | **0.713** |
>   | 15    | 0.698   | 0.693   | **0.710** | 0.706   | **0.709**   | 0.703      | **0.712** |
>   | 30    | 0.705   | 0.704   | **0.713** | 0.663   | **0.710**   | 0.657      | **0.712** |
>   | 31    | 0.702   | 0.701   | **0.712** | 0.670   | **0.710**   | 0.621      | **0.711** |
>
>   *Table: OWL per-Transformer block. 70% Sparsity, 24.55 Perplexity*
>
>   | Layer | q\_proj | k\_proj | v\_proj | o\_proj | gate\_proj | down\_proj | up\_proj |
>   |-------|---------|---------|---------|---------|------------|------------|----------|
>   | 1     | 0.613   | 0.613   | 0.613   | 0.613   | 0.613      | 0.613      | 0.613    |
>   | 2     | 0.641   | 0.641   | 0.641   | 0.641   | 0.641      | 0.641      | 0.641    |
>   | 5     | 0.608   | 0.608   | 0.608   | 0.608   | 0.608      | 0.608      | 0.608    |
>   | 30    | 0.760   | 0.760   | 0.760   | 0.760   | 0.760      | 0.760      | 0.760    |
>   | 31    | 0.721   | 0.721   | 0.721   | 0.721   | 0.721      | 0.721      | 0.721    |
>
>
> - We found that applying OWL in a per-layer manner leads to sub-optimal performance (Perplexity: 86.285 vs 24.55). We can see that applying OWL in per-layer manner will lead to uniform sparsity of certain layers across Transformer blocks (marked in bold), such as v_proj, gate_proj and up_proj, which might be undesirable for LLM pruning.

---

> ### Author Response · Authors · 2023-11-20
> **Response to Reviewer  njDu (2/3)**
>
> **Q2:** Unstructured sparsity may be less practical as it needs special hardware for acceleration.
>
> - Thank you for raising this great question! In order to examine if the promise of our non-uniform layerwise sparsity holds in hardware-friendly regimes, we explore OWL in two hardware-friendly regimes, including:
>
>   - **(1) N:M sparsity**: Following DominoSearch [1], we opt for mixed N:8 sparsity configuration. Instead of employing a uniform N value across all layers, we allow individual layers to possess distinct N values while maintaining the **same** parameter count. We adopt OWL to determine the optimal value of N for individual layers.
>
>     *Table: Perplexity of Mixed N:M Sparsity (N refers to non-zero weights).*
>
>     | Method         | Layerwise Sparsity | Structure | Perplexity |
>     |----------------|--------------------|-----------|------------|
>     | Wanda          | Uniform            | 4:8       | 8.57       |
>     | Wanda          | OWL                | Mixed 4:8 | **8.55**   |
>     | Wanda          | Uniform            | 3:8       | 42.56      |
>     | Wanda          | OWL                | Mixed 3:8 | **21.49**  |
>     | Wanda          | Uniform            | 2:8       | 2962.00    |
>     | Wanda          | OWL                | Mixed 2:8 | **331.37** |
>
>   - **(2) Structured pruning**: Instead of pruning weights, we follow the recent methodology introduced in LLM Pruner [2], wherein entire neurons and attention heads are removed. This action facilitates direct acceleration of pruned LLMs on GPUs or TPUs. We replace the uniform layerwise sparsity used by LLM pruner to a non-uniform layerwise sparsity using OWL.
>
>       *Table: Perplexity of Structure Pruning with LLaMA-7B on WikiText2 and PTB.*
>
>     | Dataset        | Pruning Method     | Layerwise Sparsity | 20%    | 40%    | 60%    | 80%    |
>     |----------------|--------------------|--------------------|--------|--------|--------|--------|
>     | Wikitext2      | LLM Pruner         | Uniform            | 19.09  | 30.39  | 90.017 | 1228.17|
>     | Wikitext2      | LLM Pruner         | OWL                | **18.57** | **28.65** | **76.99** | **321.64** |
>     | PTB            | LLM Pruner         | Uniform            | 29.51  | 66.90  | 192.06 | 1691.87|
>     | PTB            | LLM Pruner         | OWL                | **28.82** | **53.22** | **150.16** | **502.07** |
>
> - The ensuing results presented above corroborate the extension of OWL's advantages into more hardware-friendly regimes. Across diverse practical regimes, such as N:M sparsity and structured pruning, OWL consistently yields substantial performance enhancements over its uniform baseline. Notably, in scenarios like the 3:8 and 2:8 structured sparsity regimes, OWL demonstrates its efficacy by achieving perplexity reductions of 2x and 8x, respectively, surpassing the uniform baseline. And the OWL can also be applied to structured pruning to enhance the performance of the state-of-the-art LLM Pruner. We have updated the above results in Appendix A of our revision.
>
> **Q3:** Could the definition of LOD in section 3.2. be defined more precisely? This seems to be an important analysis target in section 3. Currently it is described by text which is somewhat hard to understand.
>
> - Thanks for your question. `LoD` essentially measures the distribution of the weight outliers of each layer. The procedure to calculate `LoD` includes two steps:
>
>
>   1. **Calculating $\mathbf{A}_{ij} =\parallel \mathbf{X}_j \parallel _{2}\cdot  \left|  \mathbf{ W_ij  } \right|  $** where $\parallel \mathbf{X}_j \parallel _{2}  $ is the L2 norm of the input that connects to weight $\mathbf{ W_ij  }$. We interpret this metric as the impact of input features $\mathbf{X}_j$ on weight $\mathbf{ W_ij  }$. Since outlier features are usually 100x larger than the rest of the features, we posit that weights with larger $\mathbf{A}$ play a more important role in preserving outliers.
>
>     2. **Measuring weight outlier ratios.** Outliers in each layer are then identified as the element in $\mathbf{A}_{ij}$ whose value is $\mathbf{M}$ times larger than the averaged value. Thus, `LoD` of a given layer is defined as:
>
>     $$
>     \texttt{LoD} = \frac{\sum_{i=1}^{C_{\texttt{out}}} \sum_{j=1}^{C_{\texttt{in}}} \mathbb{I} ( \mathbf{A_ij} > \text{mean}(\mathbf{A}) \times \mathbf{M} )}{C_{\texttt{in}} C_{\texttt{out}}}
>     $$
>
>     Where:
>     - $\sum_{i=1}^{C_{\texttt{out}}} \sum_{j=1}^{C_{\texttt{in}}}$: Iterates over each weight $\mathbf{W}_{ij}$ in the layer,
>     - $\mathbb{I}$: Indicator function, returns 1 if $\mathbf{A}_{ij}$ is greater than $\text{mean}(\mathbf{A}) \times \mathbf{M}$, else 0.
>     - $\mathbf{M}$: Hyperparameter used to control the threshold to identify weight outliers.
>     - $C_{\texttt{in}}$ and $C_{\texttt{out}}$: The input and output dimensions of the weight matrix, respectively.
>
>
>   We have updated our definition in the revised version.

---

> ### Author Response · Authors · 2023-11-20
> **Response to Reviewer  njDu (3/3)**
>
> **Q4:** While it is not the focus of this work, I am interested in how OWL would perform on other networks, e.g., ConvNet or Vision Transformer. Adding such analysis would be beneficial to understand the effect of outliers on pruning LLMs.
>
> - Thanks for your question. As you requested, we have applied OWL to ConvNeXt-Base and DeiT-Base and evaluated on ImageNet-1K. The pruning metric is Wanda and we compare OWL with uniform layerwise sparsity.
>
> - Our findings reveal that OWL enhances the accuracy of sparse DeiT models in contrast to Wanda. However, for ConvNeXt models, it seems that OWL does not necessarily bring benefits to ConvNeXt (neither increase nor degrade accuracy). Overall, it seems that the performance improvement of OWL on vision models is not as pronounced as observed in Language Model Models (LLMs).
>
> - Our hypothesis is that the phenomenon of outliers is not particularly evident in vision models. According to [3], outliers in LLMs are causally related to high-frequency tokens in pre-training data. These high-frequency tokens are more prevalent in textual datasets but are relatively scarce and challenging to identify within vision datasets. Hence, the phenomenon of outliers, crucial in OWL's effectiveness, may not be as evidently present or impactful within the domain of vision models, contributing to the differing performance improvements between LLMs and vision models.
>
>   *Table: Accuracy of Sparse ConvNeXt-Base on ImageNet-1K.*
>
>   | Sparsity | 50%   | 60%   | 70%   | 80%   |
>   |----------|-------|-------|-------|-------|
>   | Wanda    | 82.72 | 80.55 | 68.18 | 6.44  |
>   | OWL + Wanda | 82.76 | 80.53 | 68.28 | 6.32  |
>
>   *Table: Accuracy of Sparse DeiT-Base on ImageNet-1K.*
>
>   | Sparsity | 50%   | 60%   | 70%   | 80%   |
>   |----------|-------|-------|-------|-------|
>   | Wanda    | 78.23 | 71.14 | 49.20 | 6.86  |
>   | OWL + Wanda | 78.40 | 71.76 | 51.24 | 7.98  |
>
>
> **Q5:** While structured sparsity enforces a fixed sparsity ratio 50%, I think OWL could be potentially applied for this type of sparsity. Instead of a dynamic sparsity ratio, we could choose to either prune a FC layer with 2:4/4:8 sparsity or skip this FC layer entirely. I think such analysis would make this work more comprehensive.
>
> - Thanks for providing such an excellent idea. Indeed, OWL has the potential to enhance 2:4 sparsity. We follow your idea and conduct experiments to evaluate it. Specifically, we use OWL to determine which layers should be skipped as dense and setting the rest layers to 2:4 sparsity. We report the results in the following table. Clearly, OWL provides a good indicator to choose which layer to skip than other baselines such as random skipping and L1 norm of weight.
>
>     *Table: 2:4 Structured Sparsity*
>
>   | Skipping metric | no skip | 2       | 5       | 10     | 15     |
>   |-----------------|---------|---------|---------|--------|--------|
>   | Random          | 11.53   | 11.521  | 10.403  | 9.196  | 8.335  |
>   | L1 norm         | 11.53   | 11.823  | 11.171  | 10.425 | 9.261  |
>   | OWL             | 11.53   | **9.559** | **8.599** | **7.79** | **7.049** |
>
>     *Table: 4:8 Structured Sparsity*
>
>   | Skipping metric | no skip | 2       | 5       | 10     | 15     |
>   |-----------------|---------|---------|---------|--------|--------|
>   | Random          | 8.566   | 8.668   | 8.25    | 7.672  | 7.187  |
>   | L1 norm         | 8.566   | 8.709   | 8.358   | 7.98   | 7.453  |
>   | OWL             | 8.566   | **7.922** | **7.488** | **7.091** | **6.653** |
>
> **Reference**
>
>
> [1] Sun, Wei, Aojun Zhou, Sander Stuijk, Rob Wijnhoven, Andrew O. Nelson, and Henk Corporaal. "DominoSearch: Find layer-wise fine-grained N: M sparse schemes from dense neural networks." Advances in neural information processing systems 34 (2021): 20721-20732.
>
> [2] Ma, Xinyin, Gongfan Fang, and Xinchao Wang. "LLM-Pruner: On the Structural Pruning of Large Language Models." arXiv preprint arXiv:2305.11627 (2023).
>
> [3] Puccetti, Giovanni, Anna Rogers, Aleksandr Drozd, and Felice Dell'Orletta. "Outliers dimensions that disrupt transformers are driven by frequency." arXiv preprint arXiv:2205.11380 (2022).

---

> > ### Comment · Reviewer_njDu · 2023-11-21
> >
> > Thank you for the comprehensive experiments. My concerns have been addressed. Thus i maintain my positive rating. I would suggest to include the extra analysis results in the paper.

---

> ### Author Response · Authors · 2023-11-21
> **Thanks for your support!**
>
> Dear Reviewer njDu,
>
> We are really grateful for your positive comments and support. We are glad to hear that your concerns have been addressed!
> We've already included the extra analysis in the appendix. Your support means a lot to us!
>
> Best,
>
> Authors

---

### Official Review · Reviewer_H127 · 2023-10-30

**Soundness:** 3 good
**Presentation:** 3 good
**Contribution:** 3 good
**Rating:** 6
**Confidence:** 4

**Summary:**

This paper discusses the challenge of deploying large language models (LLMs) and proposes a pruning methodology called Outlier Weighed Layerwise sparsity (OWL) that incorporates non-uniform layerwise sparsity ratios based on outliers in token features. The evaluation shows OWL outperforms previous methods in terms of performance at high sparsity levels. The paper also highlights the competition among tech giants to build billion-parameter LLMs.

**Strengths:**

Outlier Weighed Layerwise sparsity (OWL) incorporates non-uniform layerwise sparsity ratios based on outliers in token features in the following way:

•	The authors first analyze the distribution of "outlier" features across layers in dense LLMs. Similar in the quantization case, they find the distribution is highly non-uniform. They then study how different pruning methods like magnitude pruning, SparseGPT and Wanda affect the retention of outliers. They find performance is strongly correlated with preserving outliers.
•	Based on these insights, the key idea of OWL is to assign a target layerwise sparsity for each layer based on the outlier ratio observed in that layer. Layers with a higher outlier ratio will be assigned a lower (less aggressive) layerwise sparsity ratio. This allows layers with more important outlier features to be pruned less.
•	To calculate the layerwise sparsity ratios, OWL first calculates the Layerwise Outlier Distribution (LOD) of feature effects on weights for each layer. This captures the outlier ratio for each layer. The target layerwise sparsity for each layer is then set to be directly proportional to the outlier ratio for that layer from the LOD. This aligns the sparsity ratio with the outlier distribution, preserving more outliers in layers where they are more prevalent.

Experiments on LLaMA and OPT models show OWL outperforms baselines like Wanda and SparseGPT significantly at high sparsity levels, e.g. 61.22 perplexity reduction at 70% sparsity. The authors argue this work highlights the importance of layerwise sparsity ratio in LLM pruning, which was previously overlooked.

**Weaknesses:**

One notable drawback of this paper lies in its exclusive focus on unstructured and element-wise sparsity, which offers limited hardware advantages. Unlike SparseGPT and Wanda, the authors have not provided any data on N:M sparsity. This omission represents a significant limitation and may reduce the paper's appeal to large language model practitioners.

As indicated in Table 3, OWL's effectiveness becomes more pronounced in smaller models, such as the 7B and 13B variants, whereas its impact appears to be less significant at the 30B and 65B levels. Additionally, even in the case of the 7B and 13B models, the "improved" perplexity scores after applying OWL remain relatively poor, typically exceeding 20, which limits their practical utility.

**Questions:**

See the two questions in the weakness part

---

> ### Author Response · Authors · 2023-11-20
> **Response to Reviewer  H127 (1/2)**
>
> We are grateful for your positive comments. We have carefully read your comments and address them systematically as below.
>
> **Q1:** One notable drawback of this paper lies in its exclusive focus on unstructured and element-wise sparsity, which offers limited hardware advantages. Unlike SparseGPT and Wanda, the authors have not provided any data on N:M sparsity. This omission represents a significant limitation and may reduce the paper's appeal to large language model practitioners.
>
> - We thank you for raising this great question! In order to examine if the promise of our non-uniform layerwise sparsity holds in hardware-friendly regimes. We explore OWL in two hardware-friendly regimes, including:
>     - **(1) N:M sparsity**: Following DominoSearch [1], we opt for mixed N:8 sparsity configuration. Instead of employing a uniform N value across all layers, we allow individual layers to possess distinct N values while maintaining the **same** parameter count. We adopt OWL to determine the optimal value of N for individual layers.
>
>         *Table: Perplexity of Mixed N:M Sparsity (N refers to non-zero weights)*
>
>       | Method         | Layerwise Sparsity | Structure  | Perplexity  |
>       |----------------|--------------------|------------|-------------|
>       | Wanda          | Uniform            | 4:8        | 8.57        |
>       | Wanda          | OWL                | Mixed 4:8  | **8.55**    |
>       |                |                    |            |             |
>       | Wanda          | Uniform            | 3:8        | 42.56       |
>       | Wanda          | OWL                | Mixed 3:8  | **21.49**   |
>       |                |                    |            |             |
>       | Wanda          | Uniform            | 2:8        | 2962.00     |
>       | Wanda          | OWL                | Mixed 2:8  | **331.37**  |
>
>     - **(2) Structured pruning**: Instead of pruning weights, we follow the recent methodology introduced in LLM Pruner [2], wherein entire neurons and attention heads are removed. This action facilitates direct acceleration of pruned LLMs on GPUs or TPUs. We replace the uniform layerwise sparsity used by LLM pruner to a non-uniform layerwise sparsity using OWL.
>
>         *Table : Perplexity of Structure Pruning with LLaMA-7B on WikiText2 and PTB*
>
>       | Dataset        | Pruning Method     | Layerwise Sparsity | 20%    | 40%    | 60%    | 80%    |
>       |----------------|--------------------|--------------------|--------|--------|--------|--------|
>       | Wikitext2      | LLM Pruner         | Uniform            | 19.09  | 30.39  | 90.017 | 1228.17|
>       | Wikitext2      | LLM Pruner         | OWL                | **18.57** | **28.65** | **76.99** | **321.64** |
>       | PTB            | LLM Pruner         | Uniform            | 29.51  | 66.90  | 192.06 | 1691.87|
>       | PTB            | LLM Pruner         | OWL                | **28.82** | **53.22** | **150.16** | **502.07** |
>
> - The ensuing results presented below corroborate the extension of OWL's advantages into more hardware-friendly regimes. Across diverse practical regimes, such as N:M sparsity and structured pruning, OWL consistently yields substantial performance enhancements over its uniform baseline. Notably, in scenarios like the 3:8 and 2:8 structured sparsity regimes, OWL demonstrates its efficacy by achieving perplexity reductions of 2x and 8x, respectively, surpassing the uniform baseline. And the OWL can also be applied to structured pruning to enhance the performance of the state-of-the-art LLM pruner.

---

> ### Author Response · Authors · 2023-11-20
> **Response to Reviewer  H127 (2/2)**
>
> **Q2:** As indicated in Table 3, OWL's effectiveness becomes more pronounced in smaller models, such as the 7B and 13B variants, whereas its impact appears to be less significant at the 30B and 65B levels. Additionally, even in the case of the 7B and 13B models, the "improved" perplexity scores after applying OWL remain relatively poor, typically exceeding 20, which limits their practical utility.
>
> - Thank you for your question. Indeed, the perplexity of our 70% sparse models tends to remain relatively high, particularly noticeable in smaller models such as 7B and 13B. This observation resonates with the commonly held belief in the pruning community: larger models tend to be more amenable to achieving high sparsity levels, while smaller models pose challenges in achieving comparable pruning efficiency [3,4]. For instance, as highlighted in [4], larger models can attain matching performance with their dense counterparts even with random pruning. This inherent difficulty in effectively pruning smaller models is also evident in prior works on LLM pruning, such as SparseGPT and Wanda, where performance degradation occurs notably when attempting high sparsity levels in 7B and 13B variants.
>
> - However, it's imperative to recognize that achieving high levels of sparsity is crucial for unstructured sparsity to yield tangible speedups on GPUs by leveraging existing sparse kernels. Notably, cutting-edge sparse kernels like Sputnik [5] and Flash-LLM [6] demonstrate performance superiority of unstructured sparsity over dense computation, only when sparsity levels reach or exceed 71% and 60%, respectively.
>
>   Our research primarily targets this critical regime and demonstrates a substantial performance enhancement in 70% sparse LLaMA-7B and LLaMA-13B models, with improvements of 61.22 and 6.81 perplexity respectively over the prior state-of-the-art. More significantly, our study unveils the importance of non-uniform layerwise sparsity in pruning LLMs, a facet that has been overlooked in prior research efforts, potentially hindering future progress if disregarded.
>
> - Moreover, we demonstrate that the perplexity gap can be significantly narrowed through very short time of fine-tuning, a commonly used technique in Pre-LLMs pruning. Specifically following the methodology of Wanda, we adopt LoRA [7] with a low rank of 8 to fine-tune our 70% sparse LLaMA-7B model generated by OWL + SparseGPT, using the C4 training dataset. Remarkably, **with just a 3-hour fine-tuning on two 40GB A100 GPUs**, we successfully reduce the perplexity from 19.49 to a reasonable level of 11.15. We anticipate achieving much lower perplexity if we conduct full fine-tuning for a longer time.
>     *Table： Fine-tuning for 3 hours can significantly reduce the perplexity gap*
>   | Method            | Model    | Sparsity | Perplexity |
>   |-------------------|----------|----------|------------|
>   | No Fine-tuning    | LLaMA-7B | 70%      | 19.49      |
>   | LoRA Fine-tuning  | LLaMA-7B | 70%      | **11.15**  |
>
> - Hence, we firmly believe that our paper does provide good contributes to the community of LLM compression, paving the way for the intriguing prospect of achieving notable speedups in sparse LLMs.
>
> **Reference**
>
> [1] Sun, Wei, Aojun Zhou, Sander Stuijk, Rob Wijnhoven, Andrew O. Nelson, and Henk Corporaal. "DominoSearch: Find layer-wise fine-grained N: M sparse schemes from dense neural networks." Advances in neural information processing systems 34 (2021): 20721-20732.
>
> [2] Ma, Xinyin, Gongfan Fang, and Xinchao Wang. "LLM-Pruner: On the Structural Pruning of Large Language Models." arXiv preprint arXiv:2305.11627 (2023).
>
> [3] Li, Zhuohan, Eric Wallace, Sheng Shen, Kevin Lin, Kurt Keutzer, Dan Klein, and Joey Gonzalez. "Train big, then compress: Rethinking model size for efficient training and inference of transformers." In International Conference on machine learning, pp. 5958-5968. PMLR, 2020.
>
> [4] Liu, S., Chen, T., Chen, X., Shen, L., Mocanu, D.C., Wang, Z. and Pechenizkiy, M., 2022. The unreasonable effectiveness of random pruning: Return of the most naive baseline for sparse training. arXiv preprint arXiv:2202.02643.
>
> [5] Gale, Trevor, Matei Zaharia, Cliff Young, and Erich Elsen. "Sparse gpu kernels for deep learning." In SC20: International Conference for High Performance Computing, Networking, Storage and Analysis, pp. 1-14. IEEE, 2020.
>
> [6] Xia, Haojun, Zhen Zheng, Yuchao Li, Donglin Zhuang, Zhongzhu Zhou, Xiafei Qiu, Yong Li, Wei Lin, and Shuaiwen Leon Song. "Flash-LLM: Enabling Cost-Effective and Highly-Efficient Large Generative Model Inference with
> Unstructured Sparsity." arXiv preprint arXiv:2309.10285 (2023).
>
> [7] Hu, E.J., Shen, Y., Wallis, P., Allen-Zhu, Z., Li, Y., Wang, S., Wang, L. and Chen, W., 2021. Lora: Low-rank adaptation of large language models. arXiv preprint arXiv:2106.09685.

---

> > ### Comment · Reviewer_H127 · 2023-11-22
> >
> > Thanks for the rebuttal. My concerns have been all addressed by additional experiments. Thus, I increase my score.

---

> > > ### Author Response · Authors · 2023-11-22
> > > **Thanks for increasing your score!**
> > >
> > > Dear Reviewer H127,
> > >
> > > Thank you for your support! We are pleased to address your concerns and greatly appreciate your valuable comments, which play a crucial role in improving our work.
> > >
> > > Best,
> > >
> > > Authors

---

### Official Review · Reviewer_qHFG · 2023-10-30

**Soundness:** 4 excellent
**Presentation:** 3 good
**Contribution:** 4 excellent
**Rating:** 8
**Confidence:** 4

**Summary:**

This paper introduces a novel LLM pruning methodology called Outlier Weighed Layerwise sparsity (OWL), which incorporates outlier detection to achieve high sparsity without sacrificing performance. The paper discusses the challenges of applying traditional network pruning techniques to LLMs and highlights the importance of non-uniform layerwise sparsity. The authors demonstrate the effectiveness of OWL through empirical evaluation across various benchmarks, showing a remarkable performance gain surpassing the state-of-the-art methods.

**Strengths:**

Recent efforts have been directed toward the application of traditional network pruning techniques to LLMs, uncovering a massive number of parameters that can be pruned without hurting performance. However, achieving higher >50% sparsity for performant LLMs remains as an open challenge.

In previous quantization literature, it is known that the distribution of token features within LLMs has a strong correlation with the emergence of outliers, defined as features exhibiting significantly greater magnitudes compared to their counterparts in feature dimensions. The paper suggests that non-uniform layerwise sparsity can yield substantially improved results by leveraging the same observation, and therefore incorporates outlier detection to achieve high sparsity without sacrificing performance.

Specifically, their new LLM pruning methodology incorporates outlier detection by introducing a novel layerwise sparsity ratio, denoted as Outlier Weighed Layerwise sparsity (OWL). OWL assigns greater emphasis to layers housing a higher prevalence of outliers, thereby facilitating more nuanced coordination between sparsity in weight matrices and the presence of outliers within the layer. This approach allows for high sparsity without sacrificing performance, as it takes into account the importance of outlier structures in LLMs.

The experimental evaluation conducted in this paper demonstrates the distinct advantages offered by OWL over previous methods. The authors show that OWL achieves a remarkable performance gain, surpassing the state-of-the-art Wanda and SparseGPT by 61.22 and 6.80 perplexity at a high sparsity level of 70%, respectively. Additionally, the paper presents experiments to evaluate the zero-shot ability of various sparse LLMs on diverse zero-shot downstream tasks with prompting, and shows that OWL consistently improves accuracy across nearly all settings, with very few exceptions.

**Weaknesses:**

A quite apparent limitation of this paper is its exclusive examination of unstructured pruning, without addressing more practically relevant forms of structured pruning, such as layerwise, attention head, or N:M weight sparsity. It remains uncertain whether the proposed layerwise sparsity ratio would maintain its relevance in the context of these alternative pruning approaches. I would very much like to see some preliminary results in that regard.

Furthermore, there is a question about the continued relevance and interest in improving pruning techniques for Large Language Models (LLMs). Recent research studies (as outlined in https://arxiv.org/abs/2307.02973) have shown that pruning tends to lag behind quantization in performance when considering similar storage constraints. In general, the industry has leaned towards quantization and low-rank compression due to the more established hardware support for these methods.

With that said, it would be intriguing if the authors considered extending their ideas to areas such as mixed-precision quantization or the exploration of adaptive layer-wise rank selection, whether in the context of compression or LoRA fine-tuning.

**Questions:**

Please see above, especially the first weakness.

---

> ### Author Response · Authors · 2023-11-20
> **Response to Reviewer  qHFG (1/2)**
>
> We are immensely grateful for recognizing the significance of our work and appreciate your highly positive comments. Your insightful comments have broadly expanded the benefits of our approach and improved the quality of our paper. We aim to address each weakness pointed out systematically below:
>
> **Q1**: A quite apparent limitation of this paper is its exclusive examination of unstructured pruning, without addressing more practically relevant forms of structured pruning, such as layerwise, attention head, or N:M weight sparsity. It remains uncertain whether the proposed layerwise sparsity ratio would maintain its relevance in the context of these alternative pruning approaches. I would very much like to see some preliminary results in that regard.
>
> - We thank you for raising this great question! In order to examine if the promise of our non-uniform layerwise sparsity holds in hardware-friendly regimes. We explore OWL in two hardware-friendly regimes, including:
>     - **(1) N:M sparsity**: Following DominoSearch [1], we opt for mixed N:8 sparsity configuration. Instead of employing a uniform N value across all layers, we allow individual layers to possess distinct N values while maintaining the **same** parameter count. We adopt OWL to determine the optimal value of N for individual layers.
>
>         *Table: Perplexity of Mixed N:M Sparsity (N refers to non-zero weights) with LLaMA-7B on WikiText.*
>
>       | Method         | Layerwise Sparsity | Structure | Perplexity |
>       |----------------|--------------------|-----------|------------|
>       | Wanda          | Uniform            | 4:8       | 8.57       |
>       | Wanda          | OWL                | Mixed 4:8 | **8.55**   |
>       |                |                    |           |            |
>       | Wanda          | Uniform            | 3:8       | 42.56      |
>       | Wanda          | OWL                | Mixed 3:8 | **21.49**  |
>       |                |                    |           |            |
>       | Wanda          | Uniform            | 2:8       | 2962.00    |
>       | Wanda          | OWL                | Mixed 2:8 | **331.37** |
>
>     - **(2) Structured pruning**: Instead of pruning weights, we follow the recent methodology introduced in LLM Pruner [2], wherein entire neurons and attention heads are removed. This action facilitates direct acceleration of pruned LLMs on GPUs or TPUs. We replace the uniform layerwise sparsity used by LLM pruner to a non-uniform layerwise sparsity using OWL.
>
>         *Table : Perplexity of Structure Pruning with LLaMA-7B on WikiText and PTB*
>
>       | Dataset        | Pruning Method     | Layerwise Sparsity | 20%    | 40%    | 60%    | 80%    |
>       |----------------|--------------------|--------------------|--------|--------|--------|--------|
>       | Wikitext      | LLM Pruner         | Uniform            | 19.09  | 30.39  | 90.017 | 1228.17|
>       | Wikitext      | LLM Pruner         | OWL                | **18.57** | **28.65** | **76.99** | **321.64** |
>       |                |                    |
>       | PTB            | LLM Pruner         | Uniform            | 29.51  | 66.90  | 192.06 | 1691.87|
>       | PTB            | LLM Pruner         | OWL                | **28.82** | **53.22** | **150.16** | **502.07** |
>
>   The ensuing results presented below corroborate the extension of OWL's advantages into more hardware-friendly regimes. Across diverse practical regimes, such as N:M sparsity and structured pruning, OWL consistently yields substantial performance enhancements. Notably, in scenarios like the 3:8 and 2:8 structured sparsity regimes, OWL demonstrates its efficacy by achieving perplexity reductions of 2x and 8x, respectively, surpassing the uniform baseline. And the OWL can also be applied to structured pruning to enhance the performance of the state-of-the-art LLM pruner. We have updated the above results in Appendix A of our revision.

---

> ### Author Response · Authors · 2023-11-20
> **Response to Reviewer  qHFG (2/2)**
>
> **Q2:** It would be intriguing if the authors considered extending their ideas to areas such as mixed-precision quantization or the exploration of adaptive layer-wise rank selection, whether in the context of compression or LoRA fine-tuning.
>
> - Thanks for sharing your excellent insights. We agree with you that the industry has leaned towards quantization and low-rank compression due to the more established hardware support for these methods. It is indeed very intriguing to explore OWL to mixed-precision and LoRA fine-tuning. Given the constraints of the limited rebuttal window, we delved into the application of OWL to mixed-precision quantization, drawing inspiration from your invaluable insights. To elucidate, we pre-defined bit-widths set at 2-bit, 3-bit, and 4-bit intervals, strategically allocating higher precision to layers with higher important scores. In this context, OWL will assign higher precision to layers exhibiting a higher number of outliers. This strategic allocation is based on the premise that such layers hold greater importance in preserving overall performance. We compare OWL with two baselines to calculate layer important scores, i.e., random and L1 norm of weights. The experimental evaluations were carried out using the LLaMA-7B.
>
>     *Table : Perplexity of Mixed-Precision Quantization with LLaMA-7B on WikiText2*
>
>   | Method            | Precision       | Perplexity   |
>   |-------------------|-----------------|--------------|
>   | Same Bit-width    | 2 Bit           | 104151.84    |
>   | Same Bit-width    | 3 Bit           | 25.82        |
>   | Same Bit-width    | 4 Bit           | **6.29**         |
>   |                   |                 |              |
>   | Select with random| Mixed 3/4 Bit   | 12.04        |
>   | Select with L1 Norm| Mixed 3/4 Bit  | 14.61        |
>   | Select with OWL   | Mixed 3/4 Bit   | **9.09**     |
>   |                   |                 |              |
>   | Select with random| Mixed 2/3/4 Bit | 11455.54     |
>   | Select with L1 Norm| Mixed 2/3/4 Bit| 13959.422    |
>   | Select with OWL   | Mixed 2/3/4 Bit | **190.28**   |
>   |                   |                 |              |
>   | Select with random| Mixed 2/4 Bit   | 14817.12     |
>   | Select with L1 Norm| Mixed 2/4 Bit  | 33670.214    |
>   | Select with OWL   | Mixed 2/4 Bit   | **7505.60**  |
>
>   The results showcase the benefits of OWL over baselines in terms of layer selection for mixed-precision quantization. OWL consistently achieves better performance than the random baseline, especially in the scenarios of low precision quantization. This result shows that OWL provides an accurate way to identify important layers of LLMs to select suitable bit-widths for each layer for desirable quantization performance.
>
> **Reference**
>
> [1] Sun, Wei, Aojun Zhou, Sander Stuijk, Rob Wijnhoven, Andrew O. Nelson, and Henk Corporaal. "DominoSearch: Find layer-wise fine-grained N: M sparse schemes from dense neural networks." Advances in neural information processing systems 34 (2021): 20721-20732.
>
> [2] Ma, Xinyin, Gongfan Fang, and Xinchao Wang. "LLM-Pruner: On the Structural Pruning of Large Language Models."

---

### Official Review · Reviewer_jZ6Q · 2023-11-03

**Soundness:** 1 poor
**Presentation:** 2 fair
**Contribution:** 1 poor
**Rating:** 3
**Confidence:** 4

**Summary:**

The paper studies the impact of non-uniform layerwise sparsity in LLM pruning. In particular, authors propose a new layerwise sparsity ratio that is designed to preserve the weights connected to the "outlier" features. At high sparsity levels, the LLMs pruned with the proposed "OWL" layerwise sparsity tend to have a better perplexity than uniform sparsity.

**Strengths:**

**Presentation**
Most figures are visually very appealing.

**Related work section**
The related work gives a concise yet informative summary of the related work in the field (although I find some of them quite unnecessary).

**Baselines**
The baseline methods for layerwise sparsity are mostly well-selected.

**Weaknesses:**

**Clarity (Medium)**
Some essential concepts are confusingly defined in the text: What exactly is $D_i$ (in LOD)? I presume that it is the ratio of "outlier-ish" weights, but how can it be over 1 in Figure 1 left?

**Conclusions from "Empirical Studies I, II, III" (Major)**
The overall motivation of the proposed algorithm seems to come from the so-called "empirical studies" in section 3.2., which attempt to connect the notion of LOD (layerwise outlier distribution) with the layerwise sparsity. It is very difficult for me, however, to follow how the conclusions they make can be logically deduced from their observations (it seems to be circular logic). In particular:
- **Empirical Study 1.** The empirical observation is that LOD is non-uniform on dense Llamas. From this fact, the paper jumps to the conclusion that "employing uniform pruning across all layers would inevitably disrupt the outlier structure in layers." I find this point very difficult to follow, unless the authors have a way to prove that the LOD itself, for some arbitrary $M$, is an informative criterion for pruning. But is it really so? Why is the ratio between the weight and the layerwise average important, from the first place? Perhaps the authors are resorting to some circular logic.
- **Empirical Study 2.** This part is again very confusing. I do not see how having higher LOD for non-zero weights is equal to "preserving more outliers." The reason is twofold: First, the outliers after pruning may not necessarily be identical to the outliers before pruning. Second, the pruning also changes the average.
- **Empirical Study 3.** I cannot find a clear logical conclusion that can made from this observation. How does this motivate the algorithm?

**Target sparsity (Major)**
The empirical evaluations are mostly made at, what I would call, the "useless" sparsity level. See, for instance, Table 3. The perplexities of the OWL-pruned models are quite high compared with the dense baselines (although smaller than baselines). I do not think anybody would really care too much about which method wins in the sparsity regime where the LLMs cannot really be practically useful. Same happens in Table 4.

A much more meaningful way to compare will be comparing the critical sparsity levels, i.e., the maximum sparsity where the performance metrics remain relatively similar to the dense model. To me, it seems likely to me that OWL is no better than uniform in this sense. In fact, in table 5, uniform works comparably and mostly better than OWL at lower sparsity levels.

**Practicality (Major)**
Uniform 50% sparsity is practically more useful than non-uniform sparsity at similar global sparsity level, because of the hardwares that support 2:4 sparsity.

**ERK (Minor)**
It is quite confusing why authors compare with ERK instead of ER. ERK is something that is used for convolutional layers. Does Llama have any?

**Globa (Minor)**
Citing Frankle & Carbin (2019) for the global threshold is simply wrong (section 5.1.). Look at their figure 1, the last row.

**Typos (Minor)**
- Sometimes, the paper says "outlier-weighted" instead of "outlier-weighed."
- Section 2, paragraph 2, 3rd line from bottom, "inefficacy" should be something else.

**Questions:**

- I do not understand the sentence "However, global pruning becomes extremely expensive and inefficacy in the context of LLM pruning as shown in Table 2." Have you compared the computational cost of sorting the parameters with the cost of computing saliency scores that typical LLM pruning algorithms use? Without an actual comparison, I am not really convinced about this point.

---

> ### Author Response · Authors · 2023-11-20
> **Response to Reviewer  jZ6Q （1/6）**
>
> We thank the reviewer for the time and effort in reviewing our paper. We are glad that the reviewer recognizes that our related work to be concise yet informative, and the baseline methods are mostly well-selected.
>
> We understand and respect your perspective on our work, and we are genuinely grateful for your critical evaluation. Your insights have shed light on areas where we can further enhance the clarity and depth of our research.
>
> While we acknowledge your concerns regarding certain aspects of our paper, we respectfully disagree with the conclusion drawn regarding the overall merit of our work. We firmly believe that our insights, methodology, and findings significantly contribute to the field, addressing the challenge of pruning LLMs to high sparsity e.g., $>$50%. We have carefully read the points you raised and address them point-by-point as below.
>
> **Q1: Clarity (Medium).** Some essential concepts are confusingly defined in the text: What exactly is (in LOD)? I presume that it is the ratio of "outlier-ish" weights, but how can it be over 1 in Figure 1 left?
>
> - In the context of LOD, $D_i$ specifically denotes the outlier ratio within layer $i$. Regarding Figure 1, we want to clarify that the values depicted are indeed represented as percentages, as explicitly stated in the legend: 'Figure 1: Layerwise Outlier Distribution (LOD) (**%**) of dense LLaMA-7B, 13B, and 30B.' We appreciate the opportunity to offer this clarification.
>
> **Q2: Empirical Study 1.** The empirical observation is that LOD is non-uniform on dense Llamas. From this fact, the paper jumps to the conclusion that "employing uniform pruning across all layers would inevitably disrupt the outlier structure in layers." I find this point very difficult to follow, unless the authors have a way to prove that the LOD itself, for some arbitrary , is an informative criterion for pruning. But is it really so? Why is the ratio between the weight and the layerwise average important, from the first place? Perhaps the authors are resorting to some circular logic.
>
> - Thanks for your comments. We would like to clarify that LOD is indeed a very informative criterion for pruning. LOD measures the outlier ratio of elements in metric $\mathbf{A} =\parallel \mathbf{X} \parallel _{2}\cdot |\mathbf{W}|$whose values are much larger than the averaged value. $\mathbf{A}$ essentially serves as the pruning metric used by Wanda [1], which is the current SOTA pruning approach for LLMs (**note that we have also clearly mentioned this fact in Section 3.2**). Therefore, **the crucial role of LOD as an informative criterion for pruning has already been exhibited by previous work.**
>
>
> - Furthermore, the significance of outliers has been extensively explored in prior literature concerning language models [2,3] and LLM quantization [4,5,6]. The pivotal role of outliers to LLMs is directly illustrated by a quotation from LLM.int8() [4]: "Setting these outlier feature dimensions to zero decreases top-1 attention softmax probability mass by more than 20% and degrades validation perplexity by 600-1000% despite them only making up about 0.1% of all input features."
>
> - In addition, our ablation study in Table 5 demonstrates that reversing the LOD metric (i.e., removing more weights in layers containing a higher concentration of outliers) significantly impairs performance (i.e., ranking as the second-worst among various pruning metrics). This further highlights the informativeness of LOD as an indicator for determining layerwise sparsity.
>
> - In summary, we hold a high level of confidence that our conclusion does not rely on circular reasoning but rather is a scientific study that draws inspiration from previous works and our conclusion is thoroughly supported with solid empirical results.

---

> > ### Comment · Reviewer_jZ6Q · 2023-11-23
> > **Response (Part 1)**
> >
> > Dear authors,
> >
> > Sorry for the late comeback. I really appreciate your detailed comments. Here are one-by-one response to your responses.
> >
> > __Clarity (Medium).__ This is now resolved; thank you for adding the formula, and pointing out the (%) symbol.
> >
> > __Empirical Study 1.__ Unfortunately, this is still not clear to me. I do understand that two facts: (1) the Wanda score is (at least) an empirically useful metric for post-training sparsity. (2) outlier channels are also important considerations in quantization literature. But what is not clear by itself is why "outlier, in terms of Wanda score" is important. The fact that Wanda score is important and activation outliers (appearing in specific channels) are important does not automatically imply that LOD outlier is by itself an important quantity. It can be a vague motivation for empirical design, but I think authors are obfuscating the boundaries between what is checked, and what is not. Curious to hear authors' thought about this, if it is not too late.
> >
> > __Empirical Study 2.__ Thank you for the response. The response makes sense.
> >
> > __Empirical Study 3.__ Thanks for the explanation. But there is one sentence that is still very questionable whether it is true: " However, some recent work such as SparseGPT and Wanda show that constraint pruning granularities like uniform layerwise pruning and output-balanced pruning are more preferable to LLM pruning." Much context hidden behind the word "preferable."  I do not believe that prior works were unaware of the non-uniform impact of sparsity on layers---instead, they avoided it for practical reasons. For instance, SparseGPT gives figures on ablations that avoid pruning specific layers, explicitly mentioning that n:m sparsity constrains make small modifications to the sparsity (Fig 7 of the paper). With this in mind, I do not see any tension between the conclusions on ConvNets and LLMs. Thus I think the claimed contributions of this paper on how it can change our perspective are exaggerated and misleading; I suggest a toning down significantly, without too much rhetorical tools that degrades the overall credibility.

---

> ### Author Response · Authors · 2023-11-20
> **Response to Reviewer  jZ6Q （2/6)**
>
> **Q3: Empirical Study 2.** I do not see how having higher LOD for non-zero weights is equal to "preserving more outliers." The reason is twofold: First, the outliers after pruning may not necessarily be identical to the outliers before pruning. Second, the pruning also changes the average.
>
> - Thank you for raising this important question. To clarify, the primary objective of Empirical Study II is to investigate the impact of the current LLM pruning technique on outliers. Hence, we anticipate that various pruning methodologies might yield distinct effects on outliers post-pruning. As you rightly mentioned, the outliers observed after pruning may not necessarily align with those observed before the pruning process. Nothing is wrong with this point.
>
> - However, your second point is indeed valid and we were also aware of this during our analysis. In order to avoid the influence of pruning on average values, we maintained consistency by utilizing the pre-pruning average value to measure the outlier ratio after pruning. To illustrate, if the average value before pruning is 0.1, we utilized this same value of 0.1 to identify outlier weights after pruning. Subsequently, the number of outliers after pruning was divided by the total number of weights in the layer (including both zero and non-zero weights) to calculate the updated outlier ratio. This approach helps avoid the impact of pruning on average values, ensuring a precise evaluation of alterations in the outlier ratio. We have made our approach more clear in our revision.
>
> **Q4: Empirical Study 3.** I cannot find a clear logical conclusion that can be made from this observation. How does this motivate the algorithm?
>
> - The goal of Empirical Study III is to study the effect of pruning granularity on LLMs, a crucial facet of network pruning. It is a common belief in pruning vision models that global pruning typically leads to better performance than other types of constraint pruning granularity such as layer-wise pruning. However, some recent work such as SparseGPT and Wanda show that constraint pruning granularities like uniform layerwise pruning and output-balanced pruning are more preferable to LLM pruning. To solve this mystery, we conduct this empirical study to thoroughly investigate the effect of different pruning granularity on pruning performance. Our investigation confirms that pruning with coarser granularity results in diminished performance. In general, we observe a consistent trend of improved perplexity as the pruning granularity becomes finer, transitioning from global layerwise sparsity to uniform layerwise sparsity at the macro level, and from output-imbalanced sparsity to output-balanced sparsity at the micro level. This finding motivates us that the variance of our non-uniform layerwise sparsity cannot be too aggressive. Consequently, as we mentioned in Section 3.3, **"Additionally, we introduce a hyperparameter $\lambda$ which constrains the layerwise sparsity to fall within a small range, i.e., $S_i \in [S-\lambda, S+\lambda]$ ... This constraint is inspired by the insights gained from “Empirical Study III” which highlight the detrimental impact of overly aggressive layerwise sparsity, akin to global pruning, on sparse LLMs."**.

---

> ### Author Response · Authors · 2023-11-20
> **Response to Reviewer  jZ6Q （3/6）**
>
> **Q5: Target sparsity (Major)** The empirical evaluations are mostly made at, what I would call, the "useless" sparsity level.  It seems likely to me that OWL is no better than uniform in this sense. In fact, in table 5, uniform works comparably and mostly better than OWL at lower sparsity levels.
>
>
>
> - Thank you for raising this great question. First of all, we respectfully disagree that the high sparsity regime is "useless" and nobody would really care too much. Conversely, in the context of unstructured sparsity (which is also the main focus on this paper and also previous work such as SparseGPT and Wanda, even though they report N:M sparsity), high level of sparsity stands as the sole regime where sparse computing can surpass the performance of dense computing, using existing sparse CUDA kernels on GPUs. For instance, state-of-the-art sparse kernels like Sputnik [7] and Flash-LLM [8] demonstrate that unstructured sparse begins to outperform dense computation only when sparsity levels reach or exceed 71% and 60%, respectively. After that, larger performance gain (i.e., >3x speedups) starts to exhibit as the sparsity level continuously increases.
>
> - In the spirit of pushing the limit of unstructured sparsity in LLMs, we introduce non-uniform layerwise sparsity for the first time in LLM pruning. This idea enables achieving 70% sparsity in sparse LLMs, thereby paving the way for the promising prospect of substantial speed enhancements. Consequently, we firmly assert that our paper should not be punished for advancing the frontier of unstructured sparsity in LLMs — an achievement we consider more a merit than a drawback.
>
> - Moreover, we demonstrate that the perplexity gap can be significantly narrowed through very short time of fine-tuning, a commonly used technique in Pre-LLMs pruning. Specifically following the methodology of Wanda, we adopt LoRA [18] with a low rank of 8 to fine-tune our 70% sparse LLaMA-7B/13B model generated by OWL + SparseGPT, using the C4 training dataset. Remarkably, **with just a 3-hour fine-tuning on 2 40GB  A100 GPUs**, we successfully reduce the perplexity of LLaMA-7B from 19.49 to a reasonable level of 11.15. With 5-hour fine-tuning on 4 40GB A100 GPUs, we can reduce the perplexity of LLaMA-13B from 14.55 to 9.0. We anticipate achieving much lower perplexity if we conduct full fine-tuning for a longer time.
>
>     *Table: Fine-tuning for a short time can significantly reduce the perplexity gap*
>
>     | Method           | Model     | Sparsity | Perplexity |
>     |------------------|-----------|----------|------------|
>     | No Fine-tuning   | LLaMA-7B  | 70%      | 19.49      |
>     | LoRA Fine-tuning | LLaMA-7B  | 70%      | **11.15**  |
>     | | | | |
>     | No Fine-tuning   | LLaMA-13B  | 70%      | 14.55      |
>     | LoRA Fine-tuning | LLaMA-13B   | 70%      | **9.0**  |
>
> - In addition, we also cannot agree that Uniform work mostly better than OWL at lower sparsity levels. The observed perplexity variance between Uniform and OWL remains negligible (e.g., 0.01 - 0.02 perplexity) for sparsity levels ranging from 10% to 40%. This minute difference, to be fair, does not substantiate a clear preference for either methodology. Conversely, as the sparsity level progresses beyond 50\%, OWL significantly outperforms Uniform layerwise sparsity, thus demonstrating its pronounced efficacy in comparison.
>
>     *Table: WikiText Validation Perplexity of LLaMA-7B with Various Layerwise Sparsity Using Wanda*
>
>     | Sparsity/Perplexity | 10%   | 20%   | 30%   | 40%   | 50%   | 60%   | 70%    | 80%     |
>     |---------------------|-------|-------|-------|-------|-------|-------|--------|---------|
>     | Uniform             | **5.69** | 5.81  | **5.99** | **6.38** | 7.26  | 10.70  | 85.77   | 3499.88 |
>     | OWL (ours)          | 5.70  | **5.80** | 6.01  | 6.39  | **7.22** | **9.35** | **24.54** | **1002.8** |

---

> > ### Comment · Reviewer_jZ6Q · 2023-11-23
> > **Response (Part 2)**
> >
> > __Target Sparsity.__ I greatly appreciate your response, and especially the additional experiments. Regarding your response, I have various comments.
> > - On "high sparsity as a sole regime that has gain." This is a good point. But at the same time, is a counter-argument against unstructured sparsity in general. The speedup happens only when there is a high sparsity, only when there is a significant performance degradation. Sounds like a eulogy for any sparsity other than 2:4.
> > - Also, there is more nuance to the statement "speedup over ~70%," because that holds for each tensor and not for a global sparsity. If authors are claiming that we should count the number of tensors that we can push over 70% sparsity, it sounds like a good evaluation criterion. But without such modification, the argument does not properly support that global 70% sparsity is some threshold where the phase shifts.
> > - The "results after fine-tuning" is a very good point that I did not think of. But there is one thing missing; does the model still outperform the baselines after a short fine-tuning? Useful after fine-tuning + Outperform before fine-tuning does not lead to a conclusion that "outperform after fine-tuning" holds.
> > - At lower sparsity levels, OWL still seems meritless over Uniform. At high sparsity levels, the perplexity drop, even when it is small, results in big degradations in per-task performances (see, e.g., "Compressing LLMs: The Truth is Rarely Pure and Never Simple").

---

> ### Author Response · Authors · 2023-11-20
> **Response to Reviewer  jZ6Q （4/6）**
>
> **Q6: Practicality (Major)** Uniform 50% sparsity is practically more useful than non-uniform sparsity at similar global sparsity level, because of the hardwares that support 2:4 sparsity.
>
> - Yes, we totally agree with you that structured 2:4 sparsity is very useful in practice where uniform layerwise sparsity naturally fits this regime. To assess the potential of our non-uniform layerwise sparsity in hardware-friendly scenarios, we investigate OWL across three distinct hardware-friendly regimes:
>
>     - **(1) N:M sparsity**: Following DominoSearch [9], we opt for mixed N:8 sparsity configuration. Instead of employing a uniform N value across all layers, we allow individual layers to possess distinct N values while maintaining the **same** parameter count. We adopt OWL to determine the optimal value of N for individual layers.
>
>        *Table: Perplexity of Mixed N:M Sparsity (N refers to non-zero weights)*
>
>       | Method         | Laywewise Sparsity | Structure | Perplexity |
>       |----------------|--------------------|-----------|------------|
>       | Wanda          | Uniform            | 4:8       | 8.57       |
>       | Wanda          | OWL                | Mixed 4:8 | **8.55**   |
>       |                |                    |           |            |
>       | Wanda          | Uniform            | 3:8       | 42.56      |
>       | Wanda          | OWL                | Mixed 3:8 | **21.49**  |
>       |                |                    |           |            |
>       | Wanda          | Uniform            | 2:8       | 2962.00    |
>       | Wanda          | OWL                | Mixed 2:8 | **331.37** |
>
>     - **(2) Structured pruning**: Instead of pruning weights, we follow the recent methodology introduced in LLM Pruner [16], wherein entire neurons and attention heads are removed. This action facilitates direct acceleration of pruned LLMs on GPUs or TPUs. We replace the uniform layerwise sparsity used by LLM pruner to a non-uniform layerwise sparsity using OWL.
>
>         *Table: Perplexity of Structure Pruning with LLaMA-7B on WikiText2 and PTB.*
>
>       | Dataset    | Pruning Method | Layerwise Sparsity | 20%    | 40%    | 60%    | 80%    |
>       |------------|----------------|--------------------|--------|--------|--------|--------|
>       | Wikitext2  | LLM Pruner     | Uniform            | 19.09  | 30.39  | 90.017 | 1228.17|
>       | Wikitext2  | LLM Pruner     | OWL                | **18.57** | **28.65** | **76.99** | **321.64** |
>       | PTB        | LLM Pruner     | Uniform            | 29.51  | 66.90  | 192.06 | 1691.87|
>       | PTB        | LLM Pruner     | OWL                | **28.82** | **53.22** | **150.16** | **502.07** |
>
>     - **(3) Mixed-precision quantization**: Leveraging our non-uniform layerwise sparsity, we aim to enhance mixed-precision quantization by assigning higher precision to layers exhibiting more outliers. Following the approach outlined in [17], we utilize OWL to assign different bit precision to different layers, thereby facilitating a mixed-precision quantization strategy.Our baseline here is selecting with random and L1 norm of weights.
>
>         *Table : Perplexity of Mixed-Precision Quantization with LLaMA-7B on WikiText2.*
>
>       | Method              | Precision       | Perplexity   |
>       |---------------------|-----------------|--------------|
>       | Same Bit-width      | 2 Bit           | 104151.84    |
>       | Same Bit-width      | 3 Bit           | 25.82        |
>       | Same Bit-width      | 4 Bit           | 6.29         |
>       |                     |                 |              |
>       | Select with random  | Mixed 3/4 Bit   | 12.04        |
>       | Select with L1 Norm | Mixed 3/4 Bit   | 14.61        |
>       | Select with OWL     | Mixed 3/4 Bit   | **9.09**     |
>       |                     |                 |              |
>       | Select with random  | Mixed 2/3/4 Bit | 11455.54     |
>       | Select with L1 Norm | Mixed 2/3/4 Bit | 13959.422    |
>       | Select with OWL     | Mixed 2/3/4 Bit | **190.28**   |
>       |                     |                 |              |
>       | Select with random  | Mixed 2/4 Bit   | 14817.12     |
>       | Select with L1 Norm | Mixed 2/4 Bit   | 33670.214    |
>       | Select with OWL     | Mixed 2/4 Bit   | **7505.60**  |
>
>
>     The ensuing results presented above corroborate the extension of OWL's advantages into more hardware-friendly regimes. Across diverse practical regimes, such as N:M sparsity, structured pruning, and mixed-precision quantization, OWL consistently yields substantial performance enhancements. Notably, in scenarios like the 3:8 and 2:8 structured sparsity regimes, OWL demonstrates its efficacy by achieving perplexity reductions of 2x and 8x, respectively, surpassing the uniform baseline.

---

> > ### Comment · Reviewer_jZ6Q · 2023-11-23
> > **Response (part 3)**
> >
> > __Practicality.__ I appreciate your experiments on Mixed N:8. I do think this is a good point, but for now, this does not fully support the conclusions that OWL is a good way to prune. Two reasons: (1) The perplexity gap is still very small, potentially inside 1std point, in useful perplexity regime. (2) The baselines are not compared yet.
> >
> > For structured pruning / mixed precision quantization, the perplexities are reported only for the high-perplexity regime. So I do not think this properly refutes my previous point.
> >
> > The usefulness of the non-2:4 sparsity also sounds reasonable. But my concern regarding whether OWL beats baselines in less-performance-drop is still there.
> >
> > __Global.__ This was my misunderstanding---I was the one who was 'simply wrong!' Thank you for the correction. (also, I should have said figure 2, instead of 1).
> >
> > __Q10.__ Thank you for adding these. I would not say that the number is prohibitively large, but this is definitely a good point!
> >
> > __Overall.__ Two major concerns still remain. (1) The logical steps in LOD outliers, and (2) The meritless-ness of OWL in useful-performance-regime. Plus, there are several misleading parts that twists what is true for the sake of selling the results. I do think some of my concerns have been addressed, so I have raised some scores.

---

> > > ### Author Response · Authors · 2023-11-23
> > > **Follow-up Response (3)**
> > >
> > > **Practicality.**
> > > - To clarify, our approach does not compromise the performance in the low sparsity regime. The reported results concerning high perplexity regimes were obtained without employing fine-tuning. As demonstrated earlier, with a very brief duration of fine-tuning, we effectively reduce the perplexity to practical values while surpassing the baseline performance. We anticipate to see much better performance with extended fine-tuning periods.
> > >
> > > If you have any further concerns or queries, please don't hesitate to let us know. We are more than willing toaddress any additional questions you may have.

---

> ### Author Response · Authors · 2023-11-20
> **Response to Reviewer  jZ6Q （5/6）**
>
> - More importantly, we argue that the benefits of sparsity should not be constrained to 2:4 sparsity, whose speedup upper bound is 2x but requires extremely large matrix size, e.g., 20480x20480 [10]. In practical scenarios, the actual speedups often fall even lower, around 1.6x for linear layers and 1.24x for end-to-end latency, as reported by Wanda [1].
>
> - Conversely, while the support for unstructured sparsity in 'off-the-shelf' commodity GPUs/TPUs may presently be relatively restricted, it has exhibited rapid advancement over recent years. Advanced GPU kernels such as NVIDIA cuSPARSE [11], Sputnik [7], and Flash-LLM [8] have made significant progress in support for unstructured sparsity. For instance, Flash-LLM showcases notable speedups of 1.4×, 1.7×, and 2.1× compared to dense kernels like cuBLAS at sparsity levels of 70%, 80%, and 90%, respectively. Moreover, it achieves up to 2.0× end-to-end speedup with the OPT-175B model at 80% sparsity.
>
> - Moreover, unstructured sparsity has widely proven its practical relevance on non-GPU hardware, such as CPUs or customized accelerators. For instance, in the range of 70-90% high unstructured sparsity, XNNPACK [12] has already shown significant speedups over dense baselines on smartphone processors. For an unstructured sparse RNN, an FPGA accelerator in [13] achieved high acceleration and energy efficiency performance than commercial CPU/GPU, by maximizing the use of the embedded multiply resource available on the FPGA. Another notable success was recently demonstrated by DeepSparse [14] which successfully deploys large-scale BERT-level sparse models on modern Intel CPUs, obtaining 10× model size compression with < 1% accuracy drop, 10× CPU-inference speedup with < 2% drop, and 29× CPU-inference speedup with < 7.5% drop. More recently, the S4 hardware platform introduced by Moffett AI can support up to 32× acceleration [15].
>
> Our approach provides more motivation for practitioners and endeavors to explore the benefits of sparsity on modern deep learning models.
>
> **Q7: ERK (Minor)** It is quite confusing why authors compare with ERK instead of ER. ERK is something that is used for convolutional layers. Does Llama have any?
>
> - Thanks for your knowledgeable comment. Yes, ERK is originally designed for convolutional layers. For linear layers, it will automatically scale back to ER. Therefore, we actually adopt ER for this comparison. We have fixed this in our revision.
>
> **Q8: Global (Minor)** Citing Frankle  Carbin (2019) for the global threshold is simply wrong (section 5.1.). Look at their figure 1, the last row.
>
> - As indicated in Frankle and Carbin (2019), they employed uniform sparsity, referred to as 'layer-wise pruning,' for smaller models such as LeNet and Conv-2/4/6. Conversely, for larger models like VGG-19 and Resnet-18, they implemented global pruning, as documented in Section 4 on page 7 and Appendix I.1 on page 39.
>
> - Furthermore, upon reviewing Figure 1, specifically the last row, we did not identify any content related to a global threshold. The last row of Figure 1 is 'sparse networks (average of ten trials). Solid lines are winning tickets (average of five trials).' If you are referring to a different aspect, could you please specify the particular section or figure?
>
> **Q9: Typos (Minor)** Sometimes, the paper says "outlier-weighted" instead of "outlier-weighed." Section 2, paragraph 2, 3rd line from bottom, "inefficacy" should be something else.
>
> - Thanks for these great, detailed comments. We have fixed them and polished our paper in our revision.
>
> **Q10: Global pruning** I do not understand the sentence "However, global pruning becomes extremely expensive and inefficacy in the context of LLM pruning as shown in Table 2." Have you compared the computational cost of sorting the parameters with the cost of computing saliency scores that typical LLM pruning algorithms use? Without an actual comparison, I am not really convinced about this point.
>
> - Global pruning necessitates sorting all parameters across layers to establish a pruning threshold, a process that can incur significant memory usage and computational expense, especially given the substantial number of parameters in LLMs. As you requested, we compare the computing time of sorting the parameters with the cost of computing saliency scores using Wanda. As we can see, global pruning is extremely slower and takes more than 2000 times to execute than layer-wise pruning.
>
>     *Table : Computational cost (in seconds) of pruning LLaMA-7B to 70% sparsity.*
>
>     |                   | Sorting time (s) | Saliency score time (s) |
>     |-------------------|--------------|---------------------|
>     | Global Pruning    | 438.20       | 0.04                |
>     | Layer-wise Pruning| 0.21         | 0.04                |
>
> We sincerely appreciate the time and effort you've taken to participate in the review of our paper. If you have further questions, we are more than happy to discuss with you.

---

> ### Author Response · Authors · 2023-11-20
> **Response to Reviewer  jZ6Q （6/6)**
>
> **Reference**
>
>
> [1] Sun, M., Liu, Z., Bair, A. and Kolter, J.Z., 2023. A Simple and Effective Pruning Approach for Large Language Models. arXiv preprint arXiv:2306.11695.
>
> [2] Kovaleva, O., Kulshreshtha, S., Rogers, A. and Rumshisky, A., 2021. BERT busters: Outlier dimensions that disrupt transformers. arXiv preprint arXiv:2105.06990.
>
> [3] Puccetti, Giovanni, Anna Rogers, Aleksandr Drozd, and Felice Dell'Orletta. "Outliers dimensions that disrupt transformers are driven by frequency." arXiv preprint arXiv:2205.11380 (2022).
>
> [4] Dettmers, Tim, Mike Lewis, Younes Belkada, and Luke Zettlemoyer. "Llm. int8 (): 8-bit matrix multiplication for transformers at scale." arXiv preprint arXiv:2208.07339 (2022).
>
> [5] Xiao, Guangxuan, Ji Lin, Mickael Seznec, Hao Wu, Julien Demouth, and Song Han. "Smoothquant: Accurate and efficient post-training quantization for large language models." In International Conference on Machine Learning, pp. 38087-38099. PMLR, 2023.
>
> [6] Lin, Ji, Jiaming Tang, Haotian Tang, Shang Yang, Xingyu Dang, and Song Han. "AWQ: Activation-aware Weight Quantization for LLM Compression and Acceleration." arXiv preprint arXiv:2306.00978 (2023).
>
>
> [7] Gale, Trevor, Matei Zaharia, Cliff Young, and Erich Elsen. "Sparse gpu kernels for deep learning." In SC20: International Conference for High Performance Computing, Networking, Storage and Analysis, pp. 1-14. IEEE, 2020.
>
> [8] Xia, Haojun, Zhen Zheng, Yuchao Li, Donglin Zhuang, Zhongzhu Zhou, Xiafei Qiu, Yong Li, Wei Lin, and Shuaiwen Leon Song. "Flash-LLM: Enabling Cost-Effective and Highly-Efficient Large Generative Model Inference with Unstructured Sparsity." arXiv preprint arXiv:2309.10285 (2023).
>
> [9] Sun, Wei, Aojun Zhou, Sander Stuijk, Rob Wijnhoven, Andrew O. Nelson, and Henk Corporaal. "DominoSearch: Find layer-wise fine-grained N: M sparse schemes from dense neural networks." Advances in neural information processing systems 34 (2021): 20721-20732.
>
> [10] Mishra, Asit, Jorge Albericio Latorre, Jeff Pool, Darko Stosic, Dusan Stosic, Ganesh Venkatesh, Chong Yu, and Paulius Micikevicius. "Accelerating sparse deep neural networks." arXiv preprint arXiv:2104.08378 (2021).
>
>
> [11] Valero-Lara, P., Martínez-Pérez, I., Sirvent, R., Martorell, X., and Peña, A. J. (2017, September). NVIDIA GPUs scalability to solve multiple (batch) tridiagonal systems implementation of cuThomasBatch. In International Conference on Parallel Processing and Applied Mathematics (pp. 243-253). Cham: Springer International Publishing.
>
> [12] Elsen, Erich, Marat Dukhan, Trevor Gale, and Karen Simonyan. "Fast sparse convnets." In Proceedings of the IEEE/CVF conference on computer vision and pattern recognition, pp. 14629-14638. 2020.
>
> [13] Ashby, M., Baaij, C., Baldwin, P., Bastiaan, M., Bunting, O., Cairncross, A., Chalmers, C., Corrigan, L., Davis, S., van Doorn, N. and Fowler, J., 2019. Exploiting unstructured sparsity on next-generation datacenter hardware.
>
> [14] https://github.com/neuralmagic/deepsparse
>
> [15] Yen, Ian En-Hsu, Zhibin Xiao, and Dongkuan Xu. "S4: a High-sparsity, High-performance AI Accelerator." arXiv preprint arXiv:2207.08006 (2022).
>
> [16] Ma, Xinyin, Gongfan Fang, and Xinchao Wang. "LLM-Pruner: On the Structural Pruning of Large Language Models." arXiv preprint arXiv:2305.11627 (2023).
>
> [17] Tang, Chen, Kai Ouyang, Zhi Wang, Yifei Zhu, Wen Ji, Yaowei Wang, and Wenwu Zhu. "Mixed-Precision Neural Network Quantization via Learned Layer-Wise Importance." In European Conference on Computer Vision, pp. 259-275. Cham: Springer Nature Switzerland, 2022.
>
> [18] Hu, E.J., Shen, Y., Wallis, P., Allen-Zhu, Z., Li, Y., Wang, S., Wang, L. and Chen, W., 2021. Lora: Low-rank adaptation of large language models. arXiv preprint arXiv:2106.09685.

---

> ### Author Response · Authors · 2023-11-22
> **Last day reminder**
>
> Dear reviewer jZ6Q,
>
> Thanks again for your valuable time and insightful comments. We have provided detailed response to address your comments point-by-point.
>
> As the deadline for the Author/Reviewer discussion is approaching, it would be nice of you to let us know whether our response have addressed your concerns so that we can better improve our work. We are happy to provide any additional clarifications that you may need.
>
> Best regards!
>
> Authors

---

> ### Author Response · Authors · 2023-11-23
> **Follow-up Response (1)**
>
> We thank you for your timely response. We are glad that some of your concerns have been addressed. We'd like to address your remaining concerns point-by-point.
>
> **Empirical Study 1.**
>
> - The Wanda metric is essentially the  product  of the sum of input features and weight magnitude. Since the magnitude of feature outliers in LLMs is significantly larger than the others (more than 100x larger) while the magnitude of weights  does not exhibit such outliers [1], "outliers in terms of Wanda score'' are fairly the weighs that are connected to those outlier features, thus playing an crucial role to preserve outliers.
>
> - Moreover, the consistent performance enhancements observed in LOD across a wide range of domains including unstructured pruning, N:M sparsity, structured pruning, and mixed-precision quantization further solidify the importance of LOD in LLMs.
>
> **Empirical Study 3.**
> - We would like to clarify that the assertions regarding uniform pruning and output-balanced pruning are more preferable were not made by us, but were mentioned in the previous work Wanda and SparseGPT. We have conducted a more comprehensive reexamination of these concepts in a systematic manner, in the hope of providing valuable insights to the community.
>
> - Regarding Figure 7 from SparseGPT, it does not contradict our approach of non-uniform layerwise sparsity. On the contrary, it significantly supports our motivation. Quoting their statement: "later layers are more sensitive than earlier ones; skipping the last third of the model gives the best accuracy," which aligns precisely with our motivation that certain layers hold more significance than others. Notably, they also provide results of high-sparsity of N:M in Appendix D, which further implies high sparsity is not the "useless'' regime as stated by the reviewer.
>
> - As Reviewer njDu acknowledges, LOD also serves as a robust metric for selecting which layers to skip while setting the remaining layers to N:M sparsity (the same as SparseGPT explores). We leverage LOD to identify layers to skip as dense, setting the rest to 2:4 sparsity. Our reported results in the subsequent table affirm that OWL serves as an accurate indicator for choosing layers to skip compared to other baselines, such as random skipping and the L1 norm of weight. The number in the first row refers to the number of layers that are skipped.
>     *Table: 2:4 Structured Sparsity*
>
>   | Skipping metric | no skip | 2       | 5       | 10     | 15     |
>   |-----------------|---------|---------|---------|--------|--------|
>   | Random          | 11.53   | 11.521  | 10.403  | 9.196  | 8.335  |
>   | L1 norm         | 11.53   | 11.823  | 11.171  | 10.425 | 9.261  |
>   | OWL             | 11.53   | **9.559** | **8.599** | **7.79** | **7.049** |
>
>     *Table: 4:8 Structured Sparsity*
>
>   | Skipping metric | no skip | 2       | 5       | 10     | 15     |
>   |-----------------|---------|---------|---------|--------|--------|
>   | Random          | 8.566   | 8.668   | 8.25    | 7.672  | 7.187  |
>   | L1 norm         | 8.566   | 8.709   | 8.358   | 7.98   | 7.453  |
>   | OWL             | 8.566   | **7.922** | **7.488** | **7.091** | **6.653** |
>
> Overall, we believe that our conclusion is scientifically grounded, drawing inspiration from previous works and supported by robust empirical evidence.
>
> [1] Xiao, Guangxuan, Ji Lin, Mickael Seznec, Hao Wu, Julien Demouth, and Song Han. "Smoothquant: Accurate and efficient post-training quantization for large language models." In International Conference on Machine Learning, pp. 38087-38099. PMLR, 2023.

---

> ### Author Response · Authors · 2023-11-23
> **Follow-up Response (2)**
>
> **Target Sparsity.**
>  We greatly appreciate your response but respectfully disagree with your interpretation on unstructured sparsity.
>
> - ''high sparsity as a sole regime that has gain'' was referred to unstructured sparsity on GPUs solely. High sparsity is not what we exaggerated, but the a challenging yet tantalizing goal shared by the pruning community. Our intention is not to exaggerate this pursuit, but to highlight its importance. While we very much acknowledge and embrace the notable merits achieved at 50% sparsity (where OWL maintains performance without degradation), it's essential not to disregard the potential of high sparsity in achieving greater speedups. Neglecting the impact and possibilities offered by high sparsity would indirectly discredit the invaluable contributions of numerous pioneers in this field [1,2,3,4] and may impede the future progress of science.
>
> - Regarding your second point, there might be a misunderstanding. We do not assert that 70% is the specific threshold that we should target. Instead, we choose 70% as a representative sparsity levels that enable comparisons among different approaches. Nevertheless, under our 70% global sparsity, the sparsity levels of all tensors, excluding the initial one or two layers, all exceed 60%, as illustrated in our Figure 2. These levels have already demonstrated tangible speedups using the Flash-LLM GPU library [4].
> - **Results after fine-tuning.** As you requested, we provide the fine-tuning results of Wanda as well (3-hour fine-tuning on 2 40GB A100 GPUs). We confirm that our model still outperform Wanda after a short fine-tuning.
> | Method | Model    | Sparsity | Perplexity |
> |--------|----------|----------|------------|
> | Wanda  | LLaMA-7B | 70%      | 12.82      |
> | OWL    | LLaMA-7B | 70%      | **11.15**  |
>
> - We have never claimed that we are able to match the performance of dense LLMs at high sparsity levels, such as 70\%. Instead, our paper for the first time demonstrates the possibility of pruning LLMs to 70\% sparsity while maintaining reasonable performance. Furthermore, the observed perplexity drop can be effectively recovered through a brief period of fine-tuning, as demonstrated in our study.
>
>
>
> [1] Hoefler, Torsten, Dan Alistarh, Tal Ben-Nun, Nikoli Dryden, and Alexandra Peste. "Sparsity in deep learning: Pruning and growth for efficient inference and training in neural networks." The Journal of Machine Learning Research 22, no. 1 (2021): 10882-11005.
>
> [2] Evci, Utku, Trevor Gale, Jacob Menick, Pablo Samuel Castro, and Erich Elsen. "Rigging the Lottery: Making All Tickets Winners. arXiv e-prints, art." arXiv preprint arXiv:1911.11134 (2019).
>
> [3] Gale, Trevor, Matei Zaharia, Cliff Young, and Erich Elsen. "Sparse gpu kernels for deep learning." In SC20: International Conference for High Performance Computing, Networking, Storage and Analysis, pp. 1-14. IEEE, 2020
>
> [4] Xia, Haojun, Zhen Zheng, Yuchao Li, Donglin Zhuang, Zhongzhu Zhou, Xiafei Qiu, Yong Li, Wei Lin, and Shuaiwen Leon Song. "Flash-LLM: Enabling Cost-Effective and Highly-Efficient Large Generative Model Inference with Unstructured Sparsity." arXiv preprint arXiv:2309.10285 (2023).

---

### Official Review · Reviewer_nrxB · 2023-12-04

**Soundness:** 2 fair
**Presentation:** 3 good
**Contribution:** 2 fair
**Rating:** 5
**Confidence:** 5

**Summary:**

The paper introduces a non-uniform pruning method for Large Language Models (LLM), demonstrating strong language modeling and zero-shot capabilities.

**Strengths:**

1. The paper is well-written. The content is well-organized.

2. The proposed method achieves promising results under large sparsity.

**Weaknesses:**

1.While the paper addresses non-uniform-based pruning methods, the novelty appears to be constrained. The field of model compression has extensively discussed similar approaches. Besides, the authors employ a metric akin to Wanda's, with slight modifications to layerwise sparsity distribution.

2.The paper relies heavily on empirical conclusions without providing a solid theoretical foundation for the proposed method. The authors should offer theoretical proof explaining why non-uniform strategies perform well, especially when prevailing LLM pruning strategies have contrasting conclusions.

**Questions:**

1.Referring to larger weights as "outliers" might be misleading since the term is commonly associated with extreme values in a distribution that includes both larger and smaller values, which may not accurately represent the scenario in this context.

2.Certain empirical settings lack clarity. For instance, the "M" used to calculate LOD is set to 5 or 7. The rationale behind the choice of 5 and 7, and why other values fails, requires theoretical explanation.

3.While unstructured pruning can achieve higher theoretical sparsity, practical speedup may not align. Given the utilization of a non-uniform strategy, it is crucial to include results reflecting real performance on hardware, such as inference speed.

---

### Meta-Review · Area_Chair_7LNn · 2023-12-05

**Metareview:**

The submission presents a non-uniform layer-wise pruning method for Large Language Models (LLMs) and has been examined by five reviewers. While three reviewers are enthusiastic, two other expert reviewers have raised notable concerns.

Reviewer jZ6Q acknowledged the clarity and additional explanations provided by the author rebuttal. However, the reviewer remained unconvinced regarding the significance of "outlier in terms of Wanda score", suggesting that the authors might be blurring the lines between established and speculative findings. There are also concerns regarding the exaggeration of the paper's contributions and the potential misrepresentation of context; the nuanced discussion on sparsity regimes and its practical implications.  After discussion, the reviewer's overall assessment is mixed, recognizing improvements but still identifying major concerns, particularly regarding the logical steps in LOD outliers and the practical effectiveness of OWL. Reviewer nrxB, while commending the paper for its clear, well-organized content and promising results under large sparsity, also noted its lack of solid theoretical backing.

In analyzing these reviews, AC finds it evident that the submission has made commendable strides in addressing some complex aspects of LLM pruning, particularly in terms of clarity and empirical evidence. The paper's structure and presentation are praised, and the inclusion of additional experiments and formulae is noted as a positive aspect. However, there remain significant concerns about the method novelty, the writing clarity (putting the true contribution in a clear, credible context), the theoretical underpinnings, and the practical effectiveness in real-world hardware. The authors are encouraged to submit to the next venue, after addressing those concerns more thoroughly, potentially with a major rewriting.

**Justification For Why Not Higher Score:**

the reviewer remained unconvinced regarding the significance of "outlier in terms of Wanda score", suggesting that the authors might be blurring the lines between established and speculative findings. There are also concerns regarding the exaggeration of the paper's contributions and the potential misrepresentation of context; the nuanced discussion on sparsity regimes and its practical implications.  After discussion, the reviewer's overall assessment is mixed, recognizing improvements but still identifying major concerns, particularly regarding the logical steps in LOD outliers and the practical effectiveness of OWL. Reviewer nrxB, while commending the paper for its clear, well-organized content and promising results under large sparsity, also noted its lack of solid theoretical backing.

**Justification For Why Not Lower Score:**

The paper's structure and presentation are praised, and the inclusion of additional experiments and formulae is noted as a positive aspect.

---

### Decision · Program_Chairs · 2024-01-16

Reject